# Evidence of high N₂ fixation rates in the temperate Northeast Atlantic

Debany Fonseca-Batista[1,2], Xuefeng Li[1,3], Virginie Riou[4], Valérie Michotey[4], Florian Deman[1], François Fripiat[5], Sophie Guasco[4], Natacha Brion[1], Nolwenn Lemaitre[1,6,7], Manon Tonnard[6,8], Morgane Gallinari[6], Hélène Planquette[6], Frédéric Planchon[6], Géraldine Sarthou[6], Marc Elskens[1], Julie LaRoche[2], Lei Chou[3], Frank Dehairs[1]

[1] Analytical, Environmental and Geo-Chemistry, Earth System Sciences Research Group, Vrije Universiteit Brussel, 1050 Brussels, Belgium
[2] Department of Biology, Dalhousie University, Halifax, Nova Scotia, Canada B3H 4R2
[3] Service de Biogéochimie et Modélisation du Système Terre, Université Libre de Bruxelles, 1050 Brussels, Belgium
[4] Aix-Marseille Univ, Université de Toulon, CNRS, IRD, MIO, Marseille, France
[5] Max Planck Institute for Chemistry, Climate Geochemistry Department, 55128 Mainz, Germany
[6] Laboratoire des Sciences de l'Environnement MARin – CNRS UMR 6539 – Institut Universitaire Européen de la Mer, 29280 Plouzané, France
[7] Department of Earth Sciences, Institute of Geochemistry and Petrology, ETH-Zürich, 8092 Zürich, Switzerland
[8] Institute for Marine and Antarctic Studies, University of Tasmania, Hobart, TAS 7001, Australia

*Correspondence to*: Debany Fonseca P. Batista (dbatista8@hotmail.com)

**Abstract.** Diazotrophic activity and primary production (PP) were investigated along two transects (Belgica BG2014/14 and GEOVIDE cruises) off the western Iberian Margin and the Bay of Biscay in May 2014. Substantial N₂ fixation activity was observed at 8 of the 10 stations sampled, ranging overall from 81 to 384 µmol N m$^{-2}$ d$^{-1}$ (0.7 to 8.2 nmol N L$^{-1}$ d$^{-1}$), with two sites close to the Iberian Margin situated between 38.8° N and 40.7° N yielding rates reaching up to 1355 and 1533 µmol N m$^{-2}$ d$^{-1}$. Primary production was relatively lower along the Iberian Margin with rates ranging from 33 to 59 mmol C m$^{-2}$ d$^{-1}$, while it increased towards the northwest away from the Peninsula, reaching as high as 135 mmol C m$^{-2}$ d$^{-1}$. In agreement with the area-averaged Chl *a* satellite data contemporaneous with our study period, our results revealed that post-bloom conditions prevailed at most sites, while at the northwesternmost station the bloom was still ongoing. When converted to carbon uptake using Redfield stoichiometry, N₂ fixation could support 1 to 3% of daily PP in the euphotic layer at most sites, except at the two most active sites where this contribution to daily PP could reach up to 25%. At the two sites where N₂ fixation activity was the highest, the prymnesiophyte-symbiont *Candidatus* Atelocyanobacterium thalassa (UCYN-A) dominated the *nifH* sequence pool, while the remaining recovered sequences belonged to non-cyanobacterial phylotypes. At all the other sites however, the recovered *nifH* sequences were exclusively assigned phylogenetically to non-cyanobacterial phylotypes. The intense N₂ fixation activities recorded at the time of our study were likely promoted by the availability of phytoplankton-derived organic matter produced during the spring bloom, as evidenced by the significant surface particulate organic carbon concentrations. Also, the presence of excess phosphorus signature in surface waters seemed to contribute to sustaining N₂ fixation, particularly at the sites with extreme activities. These results provide a mechanistic understanding of the unexpectedly high N₂ fixation in productive waters of the temperate North Atlantic, and highlight the importance of N₂ fixation for future assessment of global N inventory.

## 1 Introduction

Dinitrogen ($N_2$) fixation is the major pathway of nitrogen (N) input to the global ocean and thereby contributes to sustaining oceanic primary productivity (Falkowski, 1997). The conversion by $N_2$-fixing micro-organisms (diazotrophs) of dissolved $N_2$ gas into bioavailable nitrogen also contributes to new production in the euphotic layer and as such, to the subsequent sequestration of atmospheric carbon dioxide into the deep ocean (Gruber, 2008). Estimating the overall contribution of $N_2$ fixation to carbon sequestration in the ocean requires an assessment of the global marine $N_2$ fixation.

Until recently most studies of $N_2$ fixation have focused on the tropical and subtropical regions of the global ocean, with few attempts to measure $N_2$ fixation at higher latitudes, with the exception of enclosed brackish seas (Ohlendieck et al., 2000; Luo et al., 2012; Farnelid et al., 2013). The intense research efforts in the low latitude regions stem for the observable presence of cyanobacterial diazotrophs such as the diatom-diazotroph association (DDA) and the colony-forming filamentous *Trichodesmium* (Capone, 1997; Capone et al., 2005; Foster et al., 2007). *Trichodesmium* in particular, was long considered as the most active diazotroph in the global ocean. It has mostly been reported in tropical and subtropical oligotrophic oceanic waters which are thought to represent the optimal environment for its growth and $N_2$-fixing activity (Dore et al., 2002; Breitbarth et al., 2007; Montoya et al., 2007; Needoba et al., 2007; Moore et al., 2009; Fernández et al., 2010; Snow et al., 2015). In low latitude regions, warm stratified surface waters depleted in dissolved inorganic nitrogen (DIN), are assumed to give a competitive advantage to diazotrophs over other phytoplankton since only they can draw N from the unlimited dissolved $N_2$ pool for their biosynthesis. As such, past estimates of global annual $N_2$ fixation were mainly based on information gathered from tropical and subtropical regions, while higher latitude areas have been poorly explored for diazotrophic activity (Luo et al., 2012).

Studies using genetic approaches targeting the *nifH* gene encoding the nitrogenase enzyme, essential for diazotrophy, have shown the presence of diverse diazotrophs throughout the world's oceans, extending their ecological niche (Farnelid et al., 2011; Cabello et al., 2015; Langlois et al., 2015). Small diazotrophs such as unicellular diazotrophic cyanobacteria (UCYN classified in groups A, B and C) and non-cyanobacterial diazotrophs, mostly heterotrophic bacteria (e.g. Alpha- and Gammaproteobacteria), have been observed over a wide range of depths and latitudes, thereby expanding the potential for diazotrophy to a much broader geographic scale (Langlois et al., 2005, 2008; Krupke et al., 2014; Cabello et al., 2015). The discovery of a methodological bias associated to the commonly used $^{15}N_2$ bubble-addition technique (Mohr et al., 2010) and the presence of an abundant diazotrophic community in high latitude regions actively fixing $N_2$ (Needoba et al., 2007; Rees et al., 2009; Blais et al., 2012; Mulholland et al., 2012; Shiozaki et al., 2015), indicate that more efforts are needed to better constrain oceanic $N_2$ fixation and diazotrophic diversity at higher latitudes.

In the Northeast Atlantic, the large input of iron-rich Saharan dust alleviating dissolved iron (dFe) limitation of the nitrogenase activity (Fe being a co-factor of the $N_2$-fixing enzyme) (Raven, 1988; Howard & Rees, 1996; Mills et al., 2004; Snow et al., 2015) and the upwelling of subsurface waters with low DIN (dissolved inorganic nitrogen) to phosphate ratios, make this region highly favorable for $N_2$ fixation activity (Deutsch et al., 2007; Moore et al., 2009). In addition, the eastern North Atlantic has been observed to harbour a highly active and particularly diverse diazotrophic community (Langlois et al., 2008; Moore et al., 2009; Großkopf et al., 2012; Ratten et al., 2015; Fonseca-Batista et al., 2017) not only in the tropical and subtropical regions but also in the temperate Iberian region which was reported to be a hotspot for the globally important prymnesiophyte-UCYN-A symbiotic associations (Cabello et al., 2015). Earlier studies in the Iberian open waters investigated diazotrophic activity either under stratified water column conditions of boreal summer and autumn (Moore et al., 2009; Benavides et al., 2011; Snow et

al., 2015; Fonseca-Batista et al., 2017) or during the winter convection period (Rijkenberg et al., 2011; Agawin et al.,
2014). Here, we present $N_2$ fixation rate measurements and the taxonomic affiliation of the diazotrophic community
from two consecutive campaigns carried out in the Northeast sector of the Atlantic Ocean in May 2014, during and
after the spring bloom.

## 2 Material and Methods

### 2.1 Site description and sample collection

Field experiments were conducted during two nearly simultaneous cruises in May 2014. The Belgica BG2014/14
cruise (21–30 May 2014, R/V *Belgica*), investigated the Bay of Biscay and the western Iberian Margin. In parallel,
the GEOVIDE expedition in the framework of the international GEOTRACES program (GA01 section, May 16 to
June 29 2014, R/V *Pourquoi pas?*) sailed from the Portuguese shelf area towards Greenland and ended in
Newfoundland, Canada (http://dx.doi.org/10.17600/14000200). $N_2$ fixation activities were determined at ten stations
within the Iberian Basin, among which four sites were investigated during the GEOVIDE cruise (stations Geo-1,
Geo-2, Geo-13 and Geo-21) and six sites during the BG2014/14 cruise (stations Bel-3, Bel-5, Bel-7, Bel-9, Bel-11
and Bel-13; Fig. 1).
All sampling sites were located within the Iberian Basin Portugal Current System (PCS) (Ambar and Fiúza, 1994)
which is influenced by highly fluctuating wind stresses (Frouin et al., 1990). The predominant upper layer water mass
in this basin is the Eastern North Atlantic Central Water (ENACW), a winter-mode water, which according to Fiúza
(1984) consists of two components (see θ/S diagrams in Supporting Information Fig. S1): (i) the lighter, relatively
warm (> 14 °C) and salty (salinity > 35.6) ENACWst formed in the subtropical Azores Front region (~35° N) when
Azores Mode Water is subducted as a result of strong evaporation and winter cooling; and (ii) the colder and less
saline ENACWsp, underlying the ENACWst, formed in the subpolar eastern North Atlantic (north of 43° N) through
winter cooling and deep convection (McCartney and Talley, 1982). The spatial distribution of these Central Waters
allowed the categorization of the sampling sites into 2 groups: (i) ENACWsp stations north of 43° N (Bel-3, Bel-5,
Bel-7, and Geo-21) only affected by the ENACWsp (Fig. S1a, b) and (ii) ENACWst stations, south of 43° N,
characterized by an upper layer influenced by the ENACWst and an subsurface layer, by the ENACWsp (Fig. S1a,
b). Most of the ENACWst stations were open ocean sites (Bel-9, Bel-11, Bel-13, and Geo-13) while two stations
were in proximity of the Iberian shelf (Geo-1 and Geo-2) (Tonnard et al., 2018).
Temperature, salinity and photosynthetically active radiation (PAR) profiles down to 1500 m depth were obtained
using a conductivity-temperature-depth (CTD) sensor (SBE 09 and SBE 911+, during the BG2014/14 and GEOVIDE
cruises, respectively) fitted to the rosette frames. For all biogeochemical measurements, seawater samples were
collected with Niskin bottles attached to the rosette and closed at specific depths in the upper 200 m. In particular, for
stable isotope incubation experiments seawater was collected in 4.5 L acid-cleaned polycarbonate (PC) bottles from
four depths corresponding to 54%, 13%, 3% and 0.2% of surface PAR at stations Bel-3, Bel-5, Bel-7, Bel-9, Bel-11,
and Geo-2. At stations Geo-1, Geo-13 and Geo-21, two additional depths corresponding to 25% and 1% of surface
PAR were also sampled for the same purpose.

### 2.2 Nutrient measurements

Ammonium ($NH_4^+$) concentrations were measured on board during both cruises, while nitrate + nitrite ($NO_3^- + NO_2^-$)
concentrations were measured on board only during the GEOVIDE expedition. During the BG2014/14 cruise,
samples for $NO_3^-$ + $NO_2^-$ and phosphate ($PO_4^{3-}$) measurements were filtered (0.2 µm) and stored at –20 °C until
analysis at the home-based laboratory. $PO_4^{3-}$ data are not available for the GEOVIDE cruise.
Nutrient concentrations were determined using the conventional fluorometric (for $NH_4^+$) (Holmes et al., 1999) and
colorimetric methods (for the other nutrients) (Grasshoff et al., 1983) with detection limits (DL) of 64 nmol $L^{-1}$
($NH_4^+$), 90 nmol $L^{-1}$ ($NO_3^-$ + $NO_2^-$) and 60 nmol $L^{-1}$ ($PO_4^{3-}$). For the BG2014/14 cruise, chlorophyll $a$ (Chl $a$)
concentrations were determined according to Yentsch and Menzel (1963). Briefly, 250 mL of seawater was filtered
onto Whatman GF/F glass microfiber filters (0.7 µm nominal pore size), followed by pigment extraction in 90%
acetone, centrifugation and fluorescence measurement using a Shimadzu RF-150 fluorometer. For the GEOVIDE
cruise, Chl $a$ concentrations were measured as described in Ras et al. (2008). Briefly, filters samples were extracted
in 100% methanol, disrupted by sonication, and clarified by vacuum filtration through Whatman GF/F filters. The
extracts were analysed by high-performance liquid chromatography (HPLC Agilent Technologies 1200).

## 2.3 $^{15}N_2$ fixation and $^{13}C$-$HCO_3^-$ uptake rates

$N_2$ fixation and primary production (PP) were determined simultaneously from the same incubation sample at each
depth in duplicate, using the $^{15}N$-$N_2$ dissolution method (Großkopf et al., 2012) and $^{13}C$-$NaHCO_3$ tracer addition
technique (Hama et al., 1983), respectively. Details concerning the applied $^{15}N_2$ dissolution method can be found in
Fonseca-Batista et al. (2017). Briefly, $^{15}N_2$-enriched seawater was prepared by degassing prefiltered (0.2 µm) low
nutrient seawater, under acid-clean conditions using a peristaltic pump slowly circulating (100 mL $min^{-1}$) the
seawater through two degassing membrane contactor systems (MiniModule, Liqui-Cel) in series, held under high
vacuum (50 mbar). The degassed water was directly transferred into 2 L gastight Tedlar bags (Sigma-Aldrich) fitted
with a septum through which 30 mL of pure $^{15}N_2$ gas (98 $^{15}N$ atom%, Eurisotop, lot number 23/051301) was injected
before the bags were shaken 24 hours for tracer equilibration. This $^{15}N_2$ gas batch was previously shown to be free of
$^{15}N$-labelled contaminants such as nitrate, nitrite, ammonium and nitrous oxide (Fonseca-Batista et al., 2017). Each
PC incubation bottle was partially filled with sampled seawater, then amended with 250 mL of $^{15}N_2$-enriched
seawater and spiked with 3 mL of $^{13}C$-labelled dissolved inorganic carbon (DIC; 200 mmol $L^{-1}$ solution of
$NaH^{13}CO_3$, 99%, Eurisotop). The $^{13}C$-DIC added to a 4.5 L incubation bottle results in a ~6.5% increment of the
initial DIC content, considered equal to the average oceanic DIC concentration (~2000 µmol $kg^{-1}$; Zeebe and Wolf-
Gladrow, 2003). This allows sufficient tracer enrichment for a sensitive detection in the particulate organic carbon
(POC) pool as a result of incorporation (Hama et al., 1983). Finally, each incubation bottle was topped off with the
original seawater sample. Samples were then incubated for 24 hours in on-deck incubators circulated with surface
seawater and wrapped with neutral density screens (Rosco) simulating the in situ irradiance conditions. After
incubation, water was transferred under helium pressure from each PC bottle into triplicate 12 mL gastight Exetainer
vials (Labco) poisoned (100 µL of saturated $HgCl_2$ solution) and pre-flushed with helium for the determination of the
$^{15}N$ and $^{13}C$ atom% enrichments of the dissolved $N_2$ (in duplicate) and DIC pools. The remaining incubated sample
was filtered onto pre-combusted MGF filters (glass microfiber filters, 0.7 µm nominal pore size, Sartorius), which
were subsequently dried at 60 °C and stored at room temperature. The natural concentration and isotopic composition
of POC and particulate nitrogen (PN) were assessed by filtering immediately after sampling an additional 4.5 L of
non-spiked seawater from each depth. All samples were measured for POC and PN concentrations and isotopic
compositions using an elemental analyser (EuroVector Euro EA 3000) coupled to an isotope ratio mass spectrometer
(Delta V Plus, Thermo Scientific) and calibrated against international certified reference materials (CRM): IAEA-N1
and IAEA-305B for N and IAEA-CH6 and IAEA-309B for C. The isotopic composition of the DIC and dissolved $N_2$
pools was determined using a gas bench system coupled to an IRMS (Nu Instruments Perspective). Exetainers vials
were first injected with He to create a 4 mL headspace and then equilibrated on a rotatory shaker: for 12 hours after
phosphoric acid addition (100 µL, 99%, Sigma-Aldrich) for DIC analyses and only for an hour without acid addition
for N$_2$ analyses. DIC measurements were corrected according to Miyajima et al. (1995) and $^{15}$N$_2$ enrichments were
calibrated with atmospheric N$_2$. N$_2$ fixation and carbon uptake volumetric rates were computed as shown in Equation

168    1:

$$N_2 \text{ or } HCO_3^- \text{ uptake } rate \ (nmol \ L^{-1}d^{-1} \text{or } \mu mol \ m^{-3}d^{-1}) = \frac{A_{PN \, or \, POC}^{final} - A_{PN \, or \, POC}^{t=0}}{A_{N_2 \, or \, DIC} - A_{PN \, or \, POC}^{t=0}} \times \frac{[PN \, or \, POC]}{\Delta t} \qquad (1)$$
where $A_{PN \, or \, POC}$ represents the $^{15}$N or $^{13}$C atom% excess of PN or POC, respectively, at the beginning (t =0) and end
(final) of the incubation, while $A_{N_2 \, or \, DIC}$ represents the $^{15}$N or $^{13}$C atom% excess of the dissolved inorganic pool (N$_2$
or DIC); and Δt represents the incubation period.
Depth-integrated rates were calculated by non-uniform gridding trapezoidal integration for each station. The DL,
defined as the minimal detectable uptake rates were determined as detailed in Fonseca-Batista et al. (2017). To do so,
the minimal acceptable $^{15}$N or $^{13}$C enrichment of PN or POC after incubation (Montoya et al., 1996) is considered to
be equal to the natural isotopic composition, specific to each sampled depth, plus three times the uncertainty obtained
for N and C isotopic analysis of CRM. All remaining experiment-specific terms are then used to recalculate the
minimum detectable uptake. Carbon uptake rates were always above their specific DL, while N$_2$ fixation was not
detectable at any of the four depths of stations Bel-3 and Bel-5, nor at Bel-9 120 m, Bel-11 45 m and Geo-21 18 m
(see Supporting Information Table S1).
**2.4 DNA sampling and *nifH* diversity analysis**
During the BG2014/14 and GEOVIDE cruises, water samples were also collected for DNA extraction and *nifH*
sequencing at the stations where N$_2$ fixation rate measurements were carried out. Two liters of seawater samples were
vacuum filtered (20 to 30 kPa) through sterile 0.2 µm 47 mm membrane filters (cellulose acetate Sartorius type 111
for BG2014/14; Millipore's Isopore - GTTP04700 for GEOVIDE) subsequently placed in cryovials directly flash
deep frozen in liquid nitrogen. At the land-based laboratory samples were transferred to a –80 °C freezer until nucleic
acid extraction.
For the BG2014/14 samples, DNA was extracted from the samples using the Power Water DNA Isolation kit
(MOBIO) and checked for integrity by agarose gel electrophoresis. The amplification of *nifH* sequences was
performed on 3–50 ng µL$^{-1}$ environmental DNA samples using one unit of Taq polymerase (5PRIME), by nested
PCR according to Zani et al. (2000) and Langlois et al. (2005). Amplicons of the predicted 359-bp size observed by
gel electrophoresis were cloned using the PGEM T Easy cloning kit (PROMEGA) according to the manufacturer's
instructions. A total of 103 clones were sequenced by the Sanger technique (GATC, Marseille).
For the GEOVIDE samples, DNA was extracted using the QIAGEN DNeasy Plant Mini Kit as instructed by the
manufacture, with a modified step to improve cell lysis. This step consisted of an incubation at 52 °C on an orbital
shaker for 1 hour (300 rpm) with 50 µL of lysozyme solution (5 mg mL$^{-1}$ in TE buffer), 45 µL of Proteinase K
solution (20 mg mL$^{-1}$ in MilliQ PCR grade water) and 400 µL of AP1 lysis buffer from the QIAGEN DNeasy Plant
Mini Kit. DNA concentration and purity were assessed with NanoDrop 2000 and then stored at -80 °C. The DNA
samples were screened for the presence of the *nifH* gene as described in Langlois et al. (2005). Samples that tested
positive were further prepared for next generation sequencing on an Illumina MiSeq platform using primers that
included the nifH1/2 primers (Langlois et al., 2005; Ratten, 2017) attached to Illumina adaptors and barcodes for
multiplexing in the Illumina MiSeq instrument. Next generation sequencing was carried out at the Integrated
Microbiome Resource (IMR) of the Centre for Comparative and Evolutionary Biology (CGEB) at Dalhousie
University (Halifax, Canada). Raw Illumina paired-end reads of *nifH* were preprocessed using the QIIME pipeline
(Quantitative Insights Into Microbial Ecology; Caporaso et al., 2010) following the IMR workflow
(https://github.com/mlangill/microbiome_helper/wiki/16S-standard-operating-procedure; Comeau et al., 2017). The
28 OTUs for the *nifH* genes presented in this study were assembled based on 96% identity of sequence reads.
DNA alignments were performed using the Molecular Evolutionary Genetics Analysis software (MEGA 7.0) (Kumar
et al., 2016) and *nifH* operational taxonomic units (*nifH*-OTUs) were defined with a maximum 5% divergence cut-
off. DNA sequences were translated into amino acid sequences, then *nifH* evolutionary distances considered as the
number of amino acid substitutions per site, were computed using the Poisson correction method (Nei, 1987). All
positions containing gaps and missing data were eliminated (see phylogenetic tree in Fig. 6). One representative
sequence of each *nifH*-OTU was deposited in GenBank under the accession numbers referenced from KY579322 to
KY579337, for the Belgica DNA samples and referenced from MH974781 to MH974795 for the GEOVIDE Iberian
samples.
**2.5 Statistical analysis**
The relationship between $N_2$ fixation activities and ambient physical and chemical properties was examined, using
SigmaPlot (Systat Software, San Jose, CA) by computing Spearman rank correlation coefficients linking depth-
integrated rates and volumetric rates of $N_2$ fixation and primary production to environmental variables. These
ambient variables were either averaged or integrated over the euphotic layer, or considered as discrete measurements.
These variables include temperature, salinity, Chl *a*, $NH_4^+$, $NO_3^- + NO_2^-$, phosphorus excess (P* = $[PO_4^{3-}]$ – $[NO_3^-$
$+NO_2^-]$ / 16) derived from in situ nutrient measurements and climatological data (Garcia et al., 2013), dissolved iron
concentrations determined for the GEOVIDE cruise (Tonnard et al., 2018) and satellite-derived dust deposition fluxes
at the time of our study (Giovanni online data system). When nutrient concentrations were below the DL we used the
DL value to run the correlation test. In addition, we ran a principal component analysis (PCA) using XLSTAT 2017
(Addinsoft, Paris, France, 2017) to get an overview of the interconnection between all the latter key variables with $N_2$
fixation at the time of our study. The output of the PCA are discussed in section 4.3.
**3 Results**
**3.1 Ambient environmental settings**
Surface waters of all the ENACWst stations showed a relatively strong stratification resulting from the progressive
spring heating, with sea surface temperature (SST) ranging from 15.3 (Geo-13) to 17.2 °C (Bel-13). At the surface,
nutrients were depleted ($NO_3^-$ + $NO_2^-$ < 0.09 µM in the upper 20 m; Fig. 2c, f) and Chl *a* concentrations were low (<
0.25 µg $L^{-1}$; Fig. 2a, d) but showed a subsurface maximum (between 0.5 and 0.75 µg $L^{-1}$ at approximately 50 m), a
common feature for oligotrophic open ocean waters. Amongst the ENACWst stations, station Geo-13 had a slightly
higher nutrient content ($NO_3^-$ + $NO_2^-$ = 0.7 µM) in the lower mixed layer depth (MLD) and a higher Chl *a*
concentration (> 0.5 µg $L^{-1}$ in the upper 35 m).
Surface waters at ENACWsp stations were less stratified (SST between 14.0 and 14.5 °C), were nutrient replete
(surface $NO_3^-$ + $NO_2^-$ ranging from 0.3 to 0.8 µM) and had a higher phytoplankton biomass (Chl *a* between 0.7 to 1.2
µg $L^{-1}$ in the upper 30 m except for station Bel-5). Highest Chl *a* values were observed at station Bel-7 (44.6° N, 9.3°
W), which appeared to be located within an anticyclonic mesoscale eddy as evidenced by the downwelling structure
detected in the Chl *a* and $NO_3^-$ + $NO_2^-$ profiles (Fig. 2a, c) at this location (as well as T and S sections, data not
shown).

## 3.2 Primary production and satellite-based Chl $a$ observations

Primary production (PP), estimated through the incorporation of enriched bicarbonate ($^{13}$C-NaHCO$_3$) into the POC pool, illustrated volumetric rates ranging from 7 to 3500 µmol C m$^{-3}$ d$^{-1}$ (see Supporting Information Table S1) and euphotic layer integrated rates ranging from 32 to 137 mmol C m$^{-2}$ d$^{-1}$ (Fig. 3a, b, and Supporting Information Table S2). PP was relatively homogenous in the Bay of Biscay (stations Bel-3, Bel-5 and Bel-7) and along the Iberian Margin (Bel-9, Bel-11, Bel-13 and Geo-1) with average rates ranging from 33 to 43 mmol C m$^{-2}$ d$^{-1}$, except for station Bel-7 where it was slightly higher (52 mmol C m$^{-2}$ d$^{-1}$; Fig. 3a, b, and Table S2), likely due to the presence of an anticyclonic mesoscale structure at this location. PP increased westwards away from the Iberian Peninsula, reaching highest values at stations Geo-13 and Geo-21 (79 and 135 mmol C m$^{-2}$ d$^{-1}$, respectively; Fig. 3b), but also slightly higher on the Portuguese shelf (reaching 59 mmol C m$^{-2}$ d$^{-1}$ at Geo-2). These results are in the range of past measurements in this region for the same period of the year, ranging from 19 to 103 mmol C m$^{-2}$ d$^{-1}$ (Marañón et al., 2000; Fernández et al., 2005; Poulton et al., 2006; Fonseca-Batista et al., 2017). Area-averaged Chl $a$ derived from satellite imagery for a time-period overlapping with ours (Giovanni online data system; Fig. 4a, b) revealed that post-bloom conditions prevailed at most sites (Bel-3 to Bel-13 and Geo-1 to Geo-13) while bloom conditions were still ongoing at station Geo-21 at the time of our study.

## 3.3 N$_2$ fixation and dominant diazotrophs at the sampling sites

Volumetric N$_2$ fixation rates were above the DL at 8 of the 10 stations sampled in this study (Bel-3 and Bel-5 being below the DL) and ranged from 0.7 to 65.4 nmol N L$^{-1}$ d$^{-1}$ (see Table S1), with areal rates ranging between 81 and 1533 µmol N m$^{-2}$ d$^{-1}$ (Fig. 3c, d, and Table S2).

We observed intense N$_2$ fixation activities at the two sites (Bel-11 and Bel-13) most affected by ENACWst (Fig. S1). At stations Bel-11 and Bel-13, volumetric rates of N$_2$ fixation ranged from 2.4 to 65.4 nmol N L$^{-1}$ d$^{-1}$, with highest rates found at surface level (65.4 and 45.0 nmol N L$^{-1}$ d$^{-1}$, respectively), while areal rates averaged 1533 and 1355 µmol N m$^{-2}$ d$^{-1}$, respectively. N$_2$ fixation was detected at all four GEOVIDE stations. Shelf-influenced (Geo-1 and Geo-2) and open ocean (Geo-13) ENACWst sites, geographically close to Bel-11 and Bel-13, also displayed high N$_2$ fixation activities with volumetric rates ranging from 1.0 to 7.1 nmol N L$^{-1}$ d$^{-1}$ (Table S1) while depth-integrated rates averaged 141, 262 and 384 µmol N m$^{-2}$ d$^{-1}$, respectively (Fig. 3c, d, and Table S2). Significant N$_2$ fixation rates were also measured at stations that exhibited the highest primary production rates, including Bel-7, Geo-13 and Geo-21 (Fig. 3). We computed the relative contribution of N$_2$ fixation to PP by converting N$_2$ fixation rates to carbon uptake using either the Redfield ratio of 6.6 or the determined median POC/PN ratio for natural particles (equivalent to the mean value of $6.3 \pm 1.1$, $\pm$ SD, n = 46; Table 1). N$_2$ fixation contributed to less than 2% of PP at the ENACWsp sites Bel-7 and Geo-21 and between 3 to 28% of PP at the ENACWst sites, except for station Bel-9 where it supported about 1% of PP.

Screening of the *nifH* genes from DNA samples collected during the BG2014/14 cruise, returned positive *nifH* presence at stations Bel-11 and Bel-13 that displayed the largest areal N$_2$ fixation rates. Cloning of the *nifH* amplicons found in surface waters (54% PAR level where volumetric rates of N$_2$ fixation were the highest) yielded 103 *nifH* sequences. No successful *nifH* amplifications were obtained at the other Belgica stations or depths where diazotrophic activities were lower or undetectable. All the clones (n = 41) recovered from station Bel-11 were taxonomically assigned to a single OTU that had 99% identity at the nucleotide level and 100% similarity at the amino acid level with the symbiotic diazotrophic cyanobacteria UCYN-A1 or *Candidatus Atelocyanobacterium thalassa*, first characterized from station ALOHA in the North Pacific (Fig. 5a and 6) (Thompson et al., 2012). While the UCYN-A OTU also dominated the clones recovered from station Bel-13, fourteen additional *nifH* phylotypes

affiliated with non-cyanobacterial diazotrophs were also recovered at that station (Fig. 5a and 6). Among these 15
OTUs, represented by a total of 62 sequenced clones, 45.2% of the sequences were affiliated to UCYN-A1 (identical
to those found at Bel-11), and the rest to heterotrophic bacteria with 25.8% affiliated to Bacteroidetes, 19.3% to
Firmicutes and 9.7% to Proteobacteria (Gamma-, Epsilon- and Delta-proteobacteria; Fig. 5a and 6). For the
GEOVIDE cruise, *nifH* screening returned positive *nifH* presence at stations Geo-2, Geo-13 and Geo-21. Next
generation sequencing of these amplicons yielded in total 21001 reads, with a range of 170 to 9239 *nifH* amplicons
per sample, belonging exclusively to non-cyanobacterial diazotrophs, with the major affiliation to Verrucomicrobia,
and Gamma-, Delta- and Alpha-proteobacteria, representing 54, 28, 15 and 1% of total *nifH* amplicons, respectively
(Fig. 5b and 6). Members of a clade that has recently been characterized from the TARA expedition through
metagenome assembled genomes of marine heterotrophic diazotrophs (Delmont et al., 2018), were found among the
Gammaproteobacteria OTU types that dominated the community at station Geo-21.
**3.4 Relationship between $N_2$ fixation rates and environmental variables**
$N_2$ fixation activities were measured in surface waters characterized by relatively low SST (12.5–17.3 °C) and a wide
range of dissolved inorganic nitrogen (DIN) concentrations ($NO_3^-$ + $NO_2^-$ from < 0.1 to 7.6 µM). Water column
integrated $N_2$ fixation tended to increase with average surface water salinity (n = 10, $p < 0.05$, Table S3) but was
inversely correlated to satellite-based dust deposition in May 2014, the month during which our sampling took place
(n = 10, $p < 0.01$). Volumetric rates of $N_2$ fixation tended to increase with temperature (n = 46, $p < 0.01$, Table S4)
and excess phosphorus concentration (only available for Belgica studied sites, n = 24, $p < 0.01$) while being
negatively correlated to nitrate plus nitrite concentration (n = 46, $p < 0.01$).
**4 Discussion**
During two quasi-simultaneous expeditions to the Iberian Basin and the Bay of Biscay in May 2014 (38.8–46.5° N),
we observed $N_2$ fixation activity in surface waters of most visited stations (except for the two northernmost sites in
the Bay of Biscay). Our results are in support of other recent studies that have observed diazotrophic communities
and significant $N_2$ fixation rates in marine environments departing from the previously established belief that
diazotrophs are preferentially associated with warm oceanic water and low fixed-nitrogen concentrations (Needoba et
al., 2007; Rees et al., 2009; Blais et al., 2012; Mulholland et al., 2012; Shiozaki et al., 2015). Although there is
growing evidence that diazotrophs and their activity can extend geographically to temperate coastal and shelf-
influenced regions, there still exist very few rate measurements at higher latitudes, especially in open waters. In the
following sections we shall (1) discuss the significance of $N_2$ fixation in the Iberian Basin as well as its relation to
primary productivity pattern and extend our view to the whole Atlantic Ocean, (2) provide information on the
taxonomic affiliation of diazotrophs present at the time of our study, and (3) explore potential environmental
conditions that may have supported this unexpectedly high diazotrophic activity in the Iberian Basin.
**4.1 Significance of $N_2$ fixation in the temperate ocean**
In the present study, we found surprisingly high $N_2$ fixation activities at most of the studied sites. Rates were
exceptionally elevated at two open ocean stations located between 38.8 and 40.7° N at about 11° W (averaging 1533
and 1355 µmol N m$^{-2}$ d$^{-1}$ at stations Bel-11 and Bel-13, respectively; Fig. 3c, d, and Tables S1 and S2). Although $N_2$
fixation was not detected in the central Bay of Biscay (stations Bel-3 and Bel-5), rates recorded at all the other sites
were relatively high, not only in shelf-influenced areas (141 and 262 µmol N m$^{-2}$ d$^{-1}$ at stations Geo-1 and Geo-2,
respectively) but also in the open ocean (average activities between 81 and 384 µmol N m$^{-2}$ d$^{-1}$ at stations Bel-7, Bel-
9, Geo-13 and Geo-21).
By fuelling the bioavailable nitrogen pool, N$_2$ fixation may support marine primary production (PP), but the extent of
this contribution needs to be established for areas outside tropical and subtropical regions. PP rates measured here are
of similar range if not slightly higher than those reported in earlier investigations in the Northeast Atlantic from
subtropical to temperate waters (32 to 137 mmol C m$^{-2}$ d$^{-1}$ relative to 19 to 103 mmol C m$^{-2}$ d$^{-1}$) (Marañón et al.,
2000; Fernández et al., 2005; Poulton et al., 2006; Fonseca-Batista et al., 2017). However, the contribution of N$_2$
fixation to PP in the present work (1–28% of PP) reached values twice as high as those reported in other studies for
the tropical and subtropical northeast Atlantic (contributions to PP ranging from < 1% to 12%) (Voss et al., 2004;
Rijkenberg et al., 2011; Fonseca-Batista et al., 2017). This observation further questions the accepted premise that
oligotrophic surface waters of tropical and subtropical regions are the key environment where diazotrophic activity
significantly supports marine primary productivity (Capone et al., 2005; Luo et al., 2014). Nevertheless, it is
important to keep in mind that our computation relies on the assumption that only photoautotrophic diazotrophs
contribute to bulk N$_2$ fixation, which may not always be the case, particularly in the present study, where mostly
heterotrophic diazotrophs were observed. However, it is likely that all the recently fixed-nitrogen ultimately becomes
available for the whole marine autotrophic community.
Previous studies in the open waters of the Iberian Basin (35–50° N, east of 25° W) reported relatively lower N$_2$
fixation rates (from < 0.1 to 140 µmol N m$^{-2}$ d$^{-1}$), regardless of whether the bubble-addition method (Montoya et al.,
1996) or the dissolution method (Mohr et al., 2010; Großkopf et al., 2012) was used. However, these studies were
carried out largely outside the bloom period, either during the late growing season (summer and autumn) (Moore et
al., 2009; Benavides et al., 2011; Snow et al., 2015; Riou et al., 2016; Fonseca-Batista et al., 2017) or during winter
(Rijkenberg et al., 2011; Agawin et al., 2014). In contrast, the present study took place in spring, during or just at the
end of the vernal phytoplankton bloom. Differences in timing of these various studies and to a lesser extent, in
methodologies (bubble-addition versus dissolution method) may explain the discrepancies in diazotrophic activity
observed between our study and earlier works. Yet, the 20 months survey by Moreira-Coello et al. (2017) in nitrogen-
rich temperate coastal waters in the southern Bay of Biscay, covering the seasonal spring bloom and upwelling
pulses, did not reveal significant N$_2$ fixation activities: from 0.1 to 1.6 µmol N m$^{-2}$ d$^{-1}$ (up to 3 orders of magnitude
lower than those reported here). However, unlike our study, this work was carried out not only using the bubble-
addition method but also in an inner coastal system, as opposed to the mainly open waters investigated here, making
it difficult to predict which variable or combination of variables caused the difference observed between the two
studies.
Our maximal values recorded at stations Bel-11 and Bel-13 are one order of magnitude higher than maximal N$_2$
fixation rates reported further south for the eastern tropical and subtropical North Atlantic (reaching up to 360–424
µmol N m$^{-2}$ d$^{-1}$) (Großkopf et al., 2012; Subramaniam et al., 2013; Fonseca-Batista et al., 2017). Besides these two
highly active sites, N$_2$ fixation rates at the other studied locations (ranging between 81 and 384 µmol N m$^{-2}$ d$^{-1}$) were
still in the upper range of values reported for the whole eastern Atlantic region. Yet, conditions favouring N$_2$ fixation
are commonly believed to be met in tropical and subtropical regions where highest activities have mostly been
measured, particularly in the eastern North Atlantic (e.g., higher seawater temperature, DIN limiting concentrations,
excess phosphorus supply through eastern boundary upwelling systems) (Capone et al., 2005; Deutsch et al., 2007;
Luo et al., 2014; Fonseca-Batista et al., 2017).
In the Atlantic Ocean, very high N$_2$ fixation rates up to ~1000 µmol N m$^{-2}$ d$^{-1}$ as observed here, have only been
reported for temperate coastal waters of the Northwest Atlantic (up to 838 µmol N m$^{-2}$ d$^{-1}$) (Mulholland et al., 2012)
and for tropical shelf-influenced and mesohaline waters of the Caribbean and Amazon River plume (maximal rates
ranging between 898 and 1600 µmol N m$^{-2}$ d$^{-1}$) (Capone et al., 2005; Montoya et al., 2007; Subramaniam et al.,
2008). Shelf and mesohaline areas have indeed been shown to harbour considerable N$_2$ fixation activity, not only in
tropical regions (Montoya et al., 2007; Subramaniam et al. 2008) but also in waters extending from temperate to polar
areas (Rees et al., 2009; Blais et al., 2012; Mulholland et al., 2012; Shiozaki et al., 2015). Yet, the environmental
conditions leading to the high N$_2$ fixation rates in these regions are currently not well understood. For tropical
mesohaline systems, the conditions proposed to drive such an intense diazotrophic activity include the occurrence of
highly competitive diatom-diazotrophs associations and the influence of excess phosphorus input (i.e., excess relative
to the canonical Redfield P/N ratio; expressed as P*) from the Amazon River (Subramaniam et al., 2008). However,
such conditions of excess P were not observed in previous studies carried out in high latitude shelf regions with
elevated N$_2$ fixation activities (Blais et al., 2012; Mulholland et al., 2012; Shiozaki et al., 2015), nor was it distinctly
apparent in the present study (see section 4.3). In addition, while tropical mesohaline regions are characterized by the
predominance of diatom-diazotroph associations (and filamentous *Trichodesmium* spp.), in temperate shelf areas the
diazotrophic community is reported to be essentially dominated by UCYN-A and heterotrophic bacteria (Rees et al.,
2009; Blais et al., 2012; Mulholland et al., 2012; Agawin et al., 2014; Shiozaki et al., 2015; Moreira-Coello et al.,

379 2017).

**4.2 Features of the diazotrophic community composition in the temperate North Atlantic**

Our qualitative assessment of *nifH* diversity revealed a predominance of UCYN-A symbionts, only at the two stations
with the highest surface N$_2$ fixation rates (up to 65.4 and 45.0 nmol N L$^{-1}$ d$^{-1}$ at Bel-11 and Bel-13, respectively;
Table S1) while the remaining *nifH* sequences recovered belonged to heterotrophic diazotrophs, at Bel-13 as well as
at all the other sites where *nifH* genes could be detected. No *Trichodesmium nifH* sequences were recovered from
either BG2014/14 or GEOVIDE DNA samples, and the absence of the filamentous cyanobacteria was also confirmed
by the CHEMTAX analysis of phytoplankton pigments (M. Tonnard, personal communication, January 2018).
Previous work in temperate regions of the global ocean, including the Iberian Margin also reported that highest N$_2$
fixation activities were predominantly related to the presence of UCYN-A symbionts, followed by heterotrophic
bacteria, while *Trichodesmium* filaments were low or undetectable (Needoba et al., 2007; Rees et al., 2009;
Mulholland et al., 2012; Agawin et al., 2014; Shiozaki et al., 2015; Moreira-Coello et al., 2017).
UCYN-A (in particular from the UCYN-A1 clade) were shown to live in symbioses with single-celled
prymnesiophyte algae (Thompson et al., 2012). This symbiotic association, considered obligate, has been reported to
be particularly abundant in the central and eastern basin of the North Atlantic (Rees et al., 2009; Krupke et al., 2014;
Cabello et al., 2015; Martínez-Pérez et al., 2016).
Besides UCYN-A, all the remaining *nifH* sequences recovered from both cruises, although obtained through different
approaches, belonged to non-cyanobacterial diazotrophs. The phylogenetic tree (Fig. 6) showed that the non-
cyanobacterial diazotrophs clustered with (1) Verrucomicrobia, a phylum yet poorly known that includes aerobic to
microaerophilic methanotrophs groups, found in a variety of environments (Khadem et al., 2010; Wertz et al., 2012),
(2) anaerobic bacteria, obligate or facultative, mostly affiliated to Cluster III phylotypes of functional nitrogenase
(e.g., Bacteriodetes, Firmicutes, Proteobacteria) and lastly (3) phylotypes from Clusters I, II, and IV (e.g.,
Proteobacteria and Firmicutes). Among the Cluster III phylotypes, Bacteroidetes are commonly encountered in the
marine environment and are known as specialized degraders of organic matter that preferably grow attached to
particles or algal cells (Fernández-Gómez et al., 2013). N$_2$ fixation activity has previously been reported in five
Bacteroidetes strains including *Bacteroides graminisolvens*, *Paludibacter propionicigenes* and *Dysgonomonas gadei*
(Inoue et al., 2015) which are the closest cultured relatives of the *nifH*-OTUs detected at station Bel-13 (Fig. 6).
Anaerobic Cluster III phylotypes have been previously recovered from different ocean basins (Church et al., 2005;
Langlois et al., 2005, 2008; Man-Aharonovich et al., 2007; Rees et al., 2009; Halm et al., 2012; Mulholland et al.,
2012). These diazotrophs were suggested to benefit from anoxic microzones found within marine snow particles or
zooplankton guts to fix $N_2$ thereby avoiding oxygenic inhibition of their nitrogenase enzyme (Braun et al., 1999;
Church et al., 2005; Scavotto et al., 2015). Therefore, the bloom to early post-bloom conditions, prevailing during our
study, were likely beneficial for the development of diazotrophic groups that depend on the availability of detrital
organic matter or on the association with grazing zooplankton. In contrast, at the northernmost Geo-21 station, we
observed a dominance of Gammaproteobacteria phylotypes belonging to a recently identified clade of marine
diazotrophs within the Oceanospirillales (Delmont et al., 2018).
These observations tend to strengthen the idea that not only UCYN-A (Cabello et al., 2015; Martínez-Pérez et al.,
2016) but also non-cyanobacterial diazotrophs (Halm et al., 2012; Shiozaki et al., 2014; Langlois et al., 2015) play a
substantial role in oceanic $N_2$ fixation. Although it is possible to assign a broad taxonomic affiliation to classify the
*nifH* genes, very little is known with respect to their physiology, their role in the ecosystem and the factors
controlling their distribution, due to the lack of representative whole genome sequences and environmentally relevant
strains available for experimentation (Bombar et al., 2016). While the widespread distribution of UCYN-A and non-
cyanobacterial diazotrophs has been reported, their contribution to in situ activity remains poorly quantified.

## 422 4.3 Key environmental drivers of $N_2$ fixation

Environmental conditions that promote autotrophic and heterotrophic $N_2$ fixation activity in the ocean are currently
not well understood (Luo et al, 2014). While heterotrophic diazotrophs would not be directly affected by the
commonly recognized environmental controls of autotrophic diazotrophy such as solar radiation, seawater
temperature and DIN, as they possess fundamentally different ecologies, the molecular and cellular processes for
sustaining $N_2$ fixation activity would nevertheless require a supply of dFe and P (Raven, 1988; Howard & Rees,
1996; Mills et al., 2004; Snow et al., 2015). Besides the need for these critical inorganic nutrients, heterotrophic $N_2$
fixation was also recently shown to be highly dependent on the availability of organic matter (Bonnet et al., 2013;
Rahav et al., 2013, 2016; Loescher et al., 2014).
Findings from the GEOVIDE cruise tend to support the hypothesis of a stimulating effect of organic matter
availability on $N_2$ fixation activity at the time of our study. Lemaitre et al. (2018) report that surface waters (upper
100–120 m) of the Iberian Basin (stations Geo-1 and Geo-13) and the West European Basin (Geo-21) carried
significant POC loads (POC of 166, 171 and 411 mmol C $m^{-2}$, respectively) with a dominant fraction of small size
POC (the 1–53 μm size fraction; 75%, 92% and 64% of the total POC, respectively). Smaller cells, usually being
slow-sinking particles, are more easily remineralized in surface waters (Villa-Alfageme et al., 2016). This is
confirmed by the very low export efficiency (only 3 to 4% of euphotic layer integrated PP) observed at stations Geo-
13 and Geo-21, suggesting an efficient shallow remineralisation (Lemaitre et al., 2018). This availability of organic
matter in the upper layers likely contributed to supplying remineralized P (organic P being generally more labile than
other organic nutrients; Vidal et al., 1999, 2003) and to enhancing the residence time of dFe originating from
atmospheric deposition due to the formation of organic ligands (Jickells, 1999; de Baar and de Jong, 2001; Sarthou et
al., 2003).
P* values from the BG2014/14 cruise (Table S1) and the climatological P* data for the Iberian Basin (Garcia et al.,
2013) do not exhibit a clear $PO_4^{3-}$ excess in the region (P* ranging from –0.1 to 0.1 μmol $L^{-1}$; Fig. 1 and Tables S1
and S2). Nevertheless, Spearman rank correlations indicate that volumetric $N_2$ fixation rates were significantly
correlated with the BG2014/14 shipboard P* values (n = 24, $p < 0.01$, Table S4), with stations Bel-11 and Bel-13
weighing heavily in this correlation. Without the data from these two sites (data not shown), the correlation between
in situ P* and $N_2$ fixation rates is no longer significant (n= 16, $p = 0.163$), while P* becomes highly correlated with
PP and Chl $a$ (n = 16, $p = 0.0257$ and 0.016, respectively). This suggests that the effect of P* on $N_2$ fixation, although
not clearly evident from absolute values, was most important at stations Bel-11 and Bel-13 but nonetheless existent at
the other sites (Bel-7 and Bel-9). The occurrence of $N_2$ fixation in oligotrophic waters displaying weak P* values,
depleted in DIN and $PO_4^{3-}$ but replete in dFe might in fact reflect the direct use by diazotrophs of dissolved organic
phosphorus (DOP). Indeed, according to Landolfi et al. (2015) diazotrophy ensures the supply of additional N and
energy for the enzymatic mineralization of DOP (synthesis of extracellular alkaline phosphatase). Therefore, a likely
enhanced DOP release towards the end of the spring bloom may have contributed to sustaining $N_2$ fixation in the
studied region. Such DOP utilization has indeed been reported for various marine organisms, particularly
diazotrophic cyanobacteria (Dyhrman et al., 2006; Dyhrman & Haley, 2006) and bacterial communities (Luo et al.,

458 2009).

Supply routes of dFe to surface waters of the investigated area relied on lateral advection from the continental shelf
(stations Geo-1 and Geo-2) (Tonnard et al., 2018), vertical mixing due to post-winter convection (Thuróczy et al.,
2010; Rijkenberg et al., 2012; García-Ibáñez et al., 2015), and/or atmospheric dust deposition (dry + wet).
Atmospheric deposition may have been particularly important for the area of stations Bel-11 and Bel-13 receiving
warm and saline surface waters from the subtropics.
Atmospheric aerosol deposition determined during the GEOVIDE cruise (Shelley et al., 2017), as well as the
satellite-based dust deposition (dry + wet) averaged over the month of May 2014 (Fig. S3b; Giovanni online satellite
data system, NASA Goddard Earth Sciences Data and Information Services Center), reveal rather weak dust loadings
over the investigated region, resulting in areal $N_2$ fixation rates being actually inversely correlated to the satellite-
based average dust input ($p < 0.01$, Table S3). In contrast, satellite-based dust deposition (dry + wet) averaged over
the month of April 2014 (i.e. preceding the timing of sampling) indicates high fluxes over the subtropical waters
located south of the studied region (Fig. S3a;). The θ/S diagrams at stations Bel-11 and Bel-13 (and to a lesser extent
at Geo-13; Fig. S1) illustrate the presence of very warm and saline waters, which were advected from the subtropics
as suggested by the satellite SST images (Fig. S2). We thus argue that advection of surface waters from south of the
study area represented a source of atmospherically derived dFe and contributed to driving the high $N_2$ fixation
activity recorded at stations Bel-11 and Bel-13. This resulted in $N_2$ fixation rates there being positively (although
weakly) correlated ($p = 0.45$, Table S3) with the April average dust input.
For the central Bay of Biscay, where $N_2$ fixation was below the DL (stations Bel-3 and Bel-5), dust deposition in
April 2014 was also the lowest, suggesting that $N_2$ fixation there might have been limited by dFe availability. Indeed,
at stations Bel-3 and Bel-5 diazotrophic activity in surface waters was boosted following dFe amendments (> 25
nmol N $L^{-1}$ $d^{-1}$; Li et al., 2018).
Thus, the enhanced $N_2$ fixation activity at stations Bel-11 and Bel-13, as compared to the other sites, was likely
stimulated by the combined effects of the presence of highly competitive prymnesiophyte-UCYN-A symbionts,
organic matter as a source of DOP, positive P* signatures and advection of subtropical surface waters enriched in
dFe.
These statements are further supported by the outcome of a multivariate statistical analysis, providing a
comprehensive view of the environmental features influencing $N_2$ fixation. A principal component analysis (PCA;
Fig. 7 and Tables S2 and S5) generated two components (or axes) explaining 68% of the system's variability. Axis 1
illustrates the productivity of the system, or more precisely the oligotrophic state towards which it was evolving. Axis
1 is defined by a strong positive relation with surface temperature (reflecting the onset of stratification, particularly
for stations Bel-11 and Bel-13; Fig. 7) and an inverse relation with PP and associated variables (Chl $a$, $NH_4^+$, $NO_3^-$ +
$NO_2^-$), which reflects the prevailing post-bloom conditions of the system. Sites characterized by a moderate (Bel-3
and Bel-5) to high (Bel-7, Geo-21 and to a lesser extent Geo-13) PP appear indeed tightly linked to these PP-
associated variables as illustrated in Fig. 7. Axis 2 is defined by the positive relation with surface salinity and P* (Fig.
7) and reflects the advection of surface waters of subtropical origin for stations Bel-11, Bel-13 and Geo-13. For
stations Geo-1 and Geo-2, the inverse relation with surface salinity (Fig. 7) is interpreted to reflect fluvial inputs
(Tonnard et al., 2018). Finally, this statistical analysis indicates that $N_2$ fixation activity was likely influenced by the
two PCA components, tentatively identified as productivity (axis 1) and surface water advection (axis 2) from the
shelf and the subtropical region.
**5 Conclusions**
The present work highlights the occurrence of elevated $N_2$ fixation activities (81–1533 µmol N m$^{-2}$ d$^{-1}$) in spring 2014
in open waters of the temperate eastern North Atlantic, off the Iberian Peninsula. These rates exceed those reported
by others for the Iberian Basin, but which were largely obtained outside the bloom period (from < 0.1 to 140 µmol N
m$^{-2}$ d$^{-1}$). In contrast, we did not detect any $N_2$ fixation activity in the central Bay of Biscay. At sites where significant
$N_2$ fixation activity was measured, rates were similar to or up to an order of magnitude larger than values reported for
the eastern tropical and subtropical North Atlantic, regions commonly believed to represent the main areas
harbouring oceanic $N_2$ fixation for the eastern Atlantic. Assuming that the carbon versus nitrogen requirements by
these $N_2$ fixers obeyed the Redfield stoichiometry, $N_2$ fixation was found to contribute 1–3% of the euphotic layer
daily PP and even up to 23–25% at the sites where $N_2$ fixation activities were the highest. The prymnesiophyte-
symbiont *Candidatus* Atelocyanobacterium thalassa (UCYN-A) contributed the most to the *nifH* sequences recovered
at the two sites where $N_2$ fixation activity were the highest, while the remaining sequences belonged exclusively to
heterotrophic bacteria. We speculate that the unexpectedly high $N_2$ fixation activity recorded at the time of our study
was sustained by (i) organic matter availability in these open waters, resulting from the prevailing vernal bloom to
post-bloom conditions, in combination with (ii) excess phosphorus signatures which appeared to be tightly related to
diazotrophic activity particularly at the two most active sites. Yet these observations and hypotheses rely on the
availability of dFe with evidence for input from shelf waters and pulsed atmospheric dust deposition being a
significant source of iron. Further studies are required to investigate this possible link between $N_2$ fixation activity
and phytoplankton bloom under iron-replete conditions in the studied region and similar environments, as these
would require to be considered in future assessment of global $N_2$ fixation.

Data availability. The data associated with the paper are available from the corresponding author upon request.

The Supplement related to this article is available.

Competing interests. The authors declare that they have no conflict of interest.

*Acknowledgements*. We thank the Captains and the crews of R/V *Belgica* and R/V *Pourquoi pas?* for their skilful
logistic support. A very special thank goes to the chief scientists G. Sarthou and P. Lherminier of the GEOVIDE
expedition for the great work experience and wonderful support on board. We would like to give special thanks to
Pierre Branellec, Michel Hamon, Catherine Kermabon, Philippe Le Bot, Stéphane Leizour, Olivier Ménage
(Laboratoire d'Océanographie Physique et Spatiale), Fabien Pérault and Emmanuel de Saint Léger (Division
Technique de l'INSU, Plouzané, France) for their technical expertise during clean CTD deployments. We thank A.
Roukaerts and D. Verstraeten for their assistance with laboratory analyses at the Vrije Universiteit Brussel. We
acknowledge Ryan Barkhouse for the collection of the DNA samples during the GEOVIDE cruise, Jennifer Tolman
and Jenni-Marie Ratten for the *nifH* amplification and Tag sequencing. P. Lherminer, P. Tréguer, E. Grossteffan, and
M. Le Goff are gratefully acknowledged for providing us with the shipboard physico-chemical data including CTD
and nitrate plus nitrite data from the GEOVIDE expedition. Shiptime for the Belgica BG2014/14 cruise was granted
by Operational Directorate 'Natural Environment' (OD Nature) of the Royal Institute of Natural Sciences, Belgium.
OD Nature (Ostend) is also acknowledged for their assistance in CTD operations and data acquisition on board the
R/V *Belgica*. This work was financed by the Flanders Research Foundation (FWO contract G0715.12N) and Vrije
Universiteit Brussel, R&D, Strategic Research Plan "Tracers of Past & Present Global Changes", and is a Belgian
contribution to SOLAS. Additional funding was provided by the Fund for Scientific Research - FNRS (F.R.S.-FNRS)
of the Wallonia-Brussels Federation (convention no. J.0150.15). X. Li was a FNRS doctorate Aspirant fellow
(mandate no. FC99216). This study was also supported, through the GEOVIDE expedition, by the French National
Research Agency (ANR-13-B506-0014), the Institut National des Sciences de L'Univers (INSU) of the Centre
National de la Recherche Scientifique (CNRS), and the French Institute for Marine Science (Ifremer). This work was
logistically supported for the by DT-INSU and GENAVIR. This publication is also a contribution to the Labex OT-
Med [ANR-11-LABEX-0061, www.otmed.fr] funded by the « Investissements d'Avenir », French Government
project of the French National Research Agency [ANR, www.agence-nationale-recherche.fr] through the A*Midex
project [ANR-11-IDEX-0001-02], funding V. Riou during the preparation of the manuscript. Finally, this work was
also supported by an NSERC Discovery grant and Ocean Frontier Institute (OFI) grant [Canada First Research
Excellence Funds] to J. LaRoche, and the OFI postdoctoral fellow D. Fonseca-Batista.

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

**Table**


834                 **Table 1:** Relative contribution (%) of $N_2$ fixation to Primary Production (PP).


| Province | Station | Latitude (° N) | Longitude (° E) | $N_2$ fixation contribution to PP (%) (Redfield 6.6 ratio) | SD | $N_2$ fixation contribution to PP (%) (mean POC/PN ratio of 6.3 ± 1.1) | SD |
|---|---|---|---|---|---|---|---|
| ENACWsp | **Bel-3** | 46.5 | -8.0 | **0** | - | **0** | - |
| | **Bel-5** | 45.3 | -8.8 | **0** | - | **0** | - |
| | **Bel-7** | 44.6 | -9.3 | **2** | 0.4 | **1** | 0.4 |
| | **Geo-21** | 46.5 | -19.7 | **1** | 0.02 | **1** | 0.0 |
| ENACWst | **Bel-9** | 42.4 | -9.7 | **1** | 0.1 | **1** | 0.1 |
| | **Bel-11** | 40.7 | -11.1 | **28** | 1.9 | **25** | 1.8 |
| | **Bel-13** | 38.8 | -11.4 | **25** | 1.3 | **23** | 1.2 |
| | **Geo-1** | 40.3 | -10.0 | **3** | 0.2 | **3** | 0.1 |
| | **Geo-2** | 40.3 | -9.5 | **3** | 0.1 | **3** | 0.1 |
| | **Geo-13** | 41.4 | -13.9 | **3** | 0.1 | **3** | 0.1 |


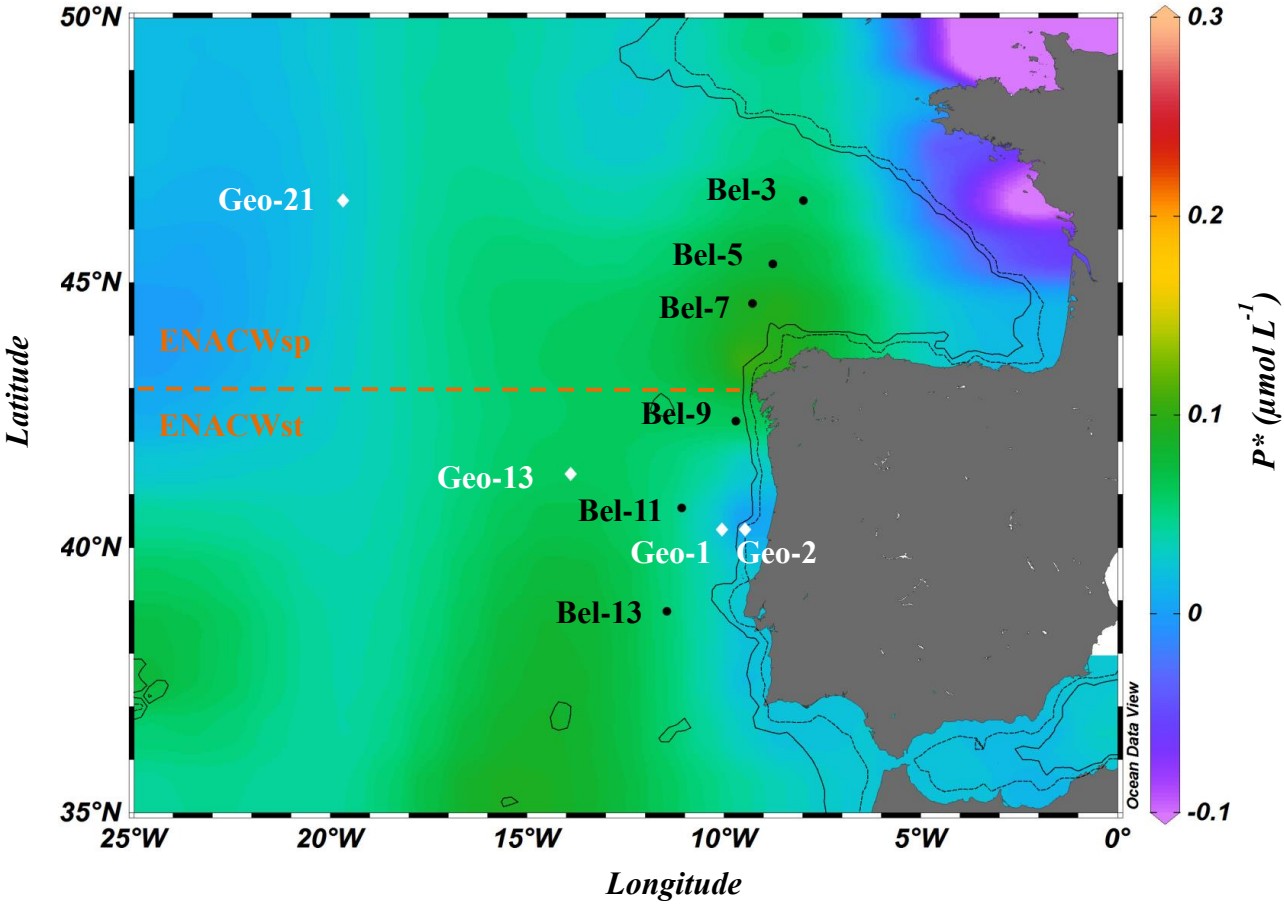


**Figure 1:** Location of sampling stations during the Belgica BG2014/14 (black labels) and GEOVIDE (white labels)
cruises (May 2014) superimposed on a map of the seasonal average phosphate excess ($P^* = [PO_4^{3-}] - [NO_3^-] / 16$) at
20 m (April to June for the period from 1955 to 2012; World Ocean Atlas 2013; Garcia et al., 2013). Areas of
dominance of the Eastern North Atlantic Central Waters of subpolar (ENACWsp) and subtropical (ENACWst) origin
are separated by a horizontal dashed line. Black dashed and solid contour lines illustrate 500 m and 1500 m isobaths,
respectively. (Schlitzer, R., Ocean Data View).


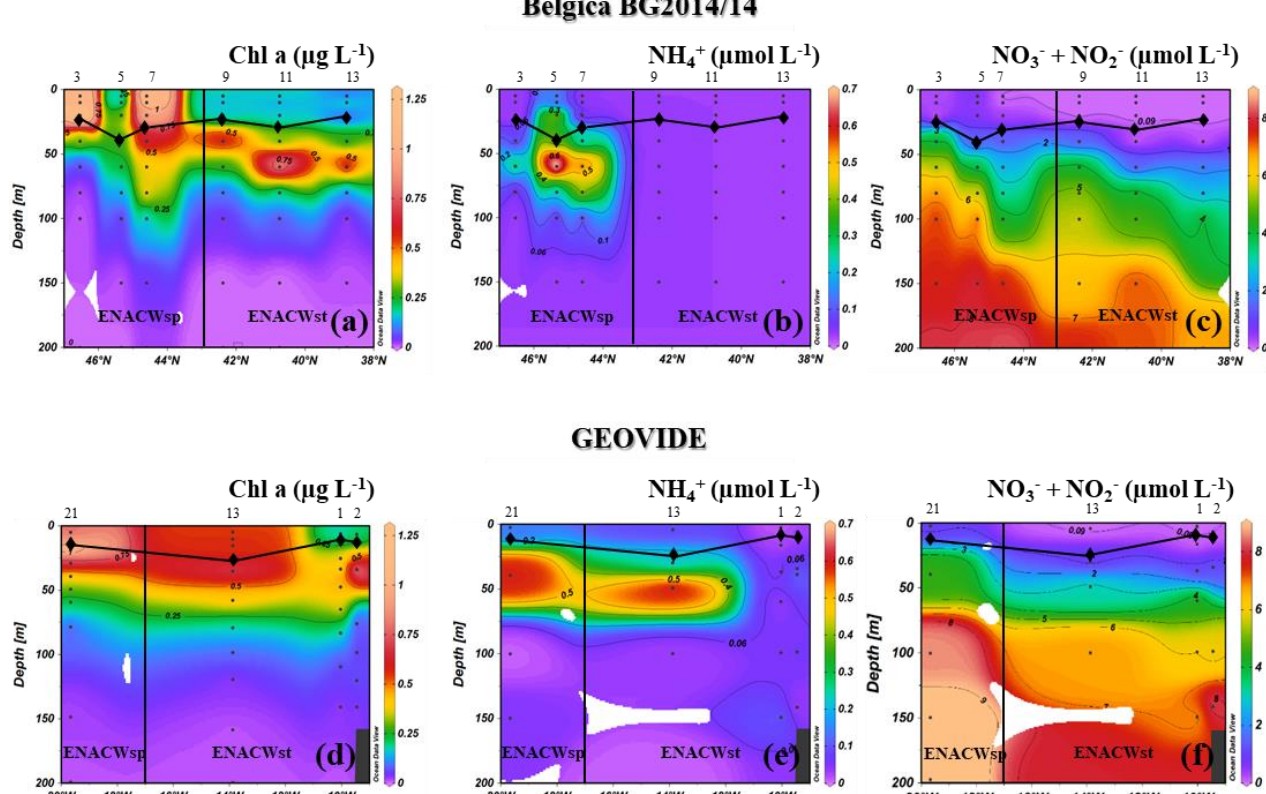

**Figure 2:** Spatial distribution of Chl *a* **(a, d)**, $NH_4^+$ **(b, e)** and $NO_3^- + NO_2^-$ **(c, f)** concentrations along the Belgica BG2014/14 **(upper panels)** and GEOVIDE **(lower panels)** cruise tracks. Station numbers are indicated above the sections. The vertical black line represents the boundary between areas with dominance of Eastern North Atlantic Waters of subpolar (ENACWsp) and subtropical (ENACWst) origin. Mixed layer depth (MLD, black lines connecting diamonds) was estimated using a temperature threshold criterion of 0.2 °C relative to the temperature at 10 m (de Boyer Montégut et al., 2004). (Schlitzer, R., Ocean Data View).

858

859

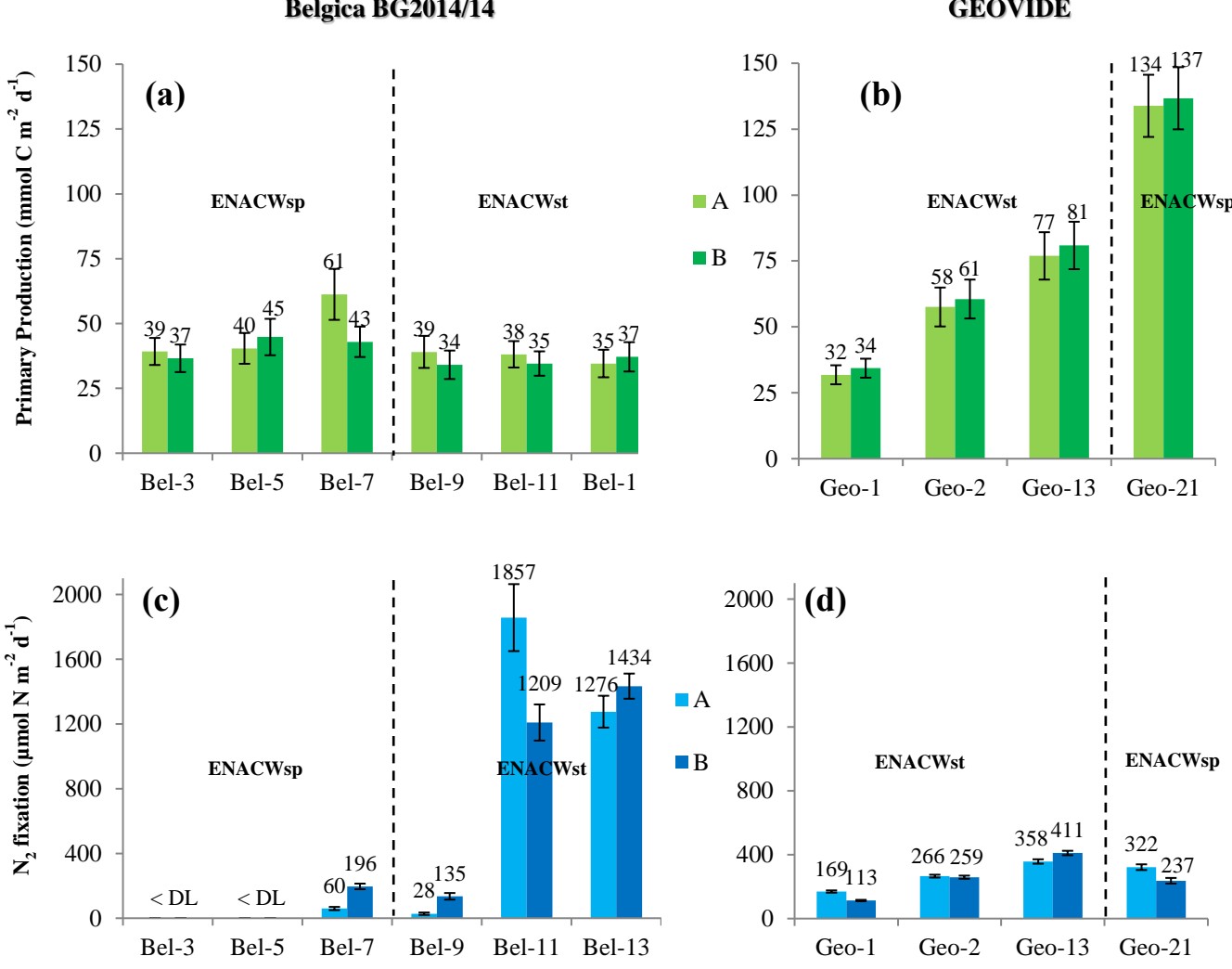

860

**Figure 3:** Spatial distribution (± SD) of depth-integrated rates of primary production **(a, b)** (duplicates are in light and dark green bars with the corresponding values in mmol C m$^{-2}$ d$^{-1}$); N$_2$ fixation **(c, d)** (duplicates are in light and dark blue bars with the corresponding values in µmol N m$^{-2}$ d$^{-1}$) determined during the Belgica BG2014/14 **(a, c)** and GEOVIDE **(b, d)** cruises. Error bars represent the propagated measurement uncertainty of all parameters used to compute volumetric uptake rates.



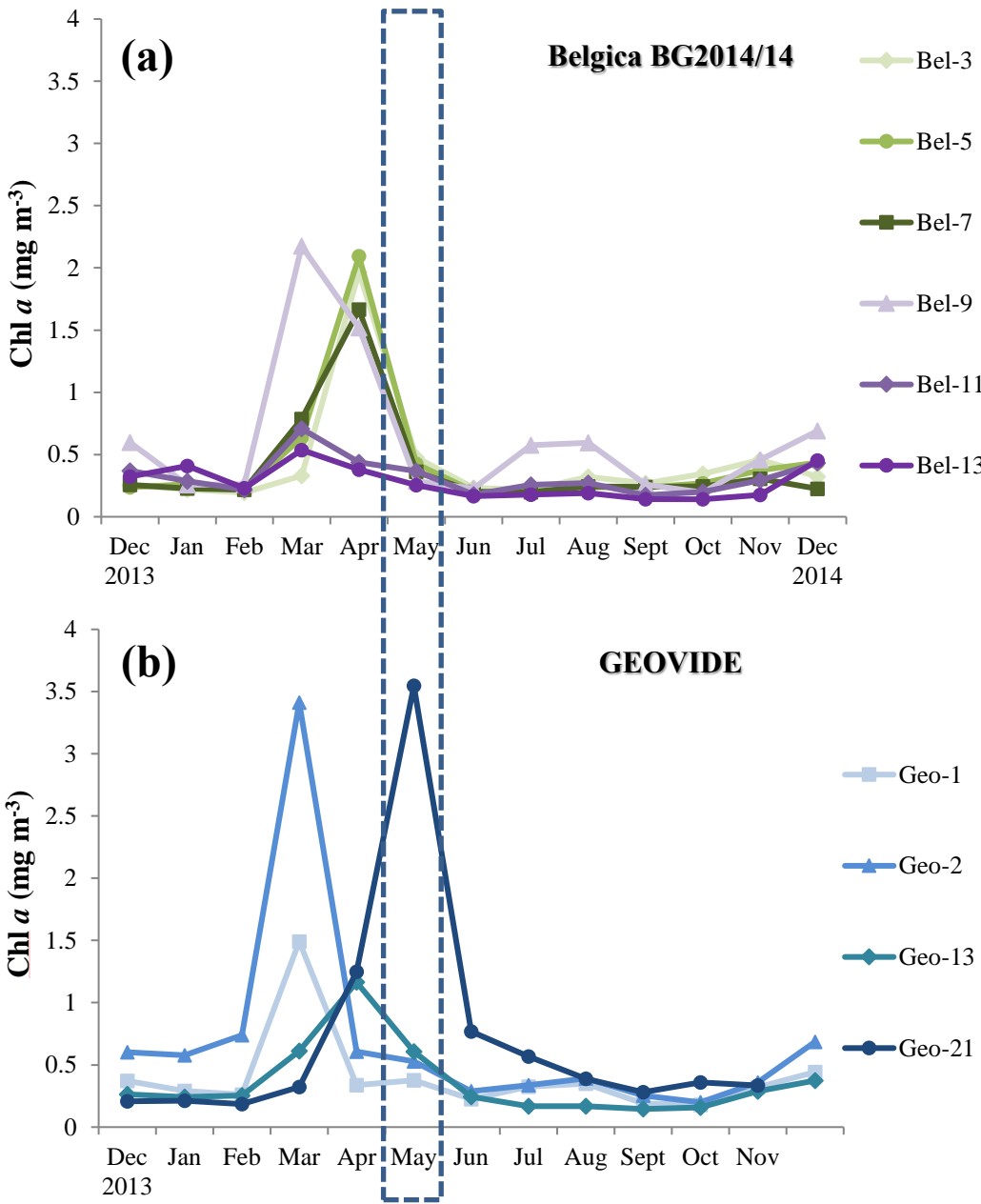


**Figure 4:** Time series of area-averaged chlorophyll a concentration (mg m$^{-3}$) registered by Aqua MODIS satellite
(Giovanni online satellite data system) between December 2013 and December 2014 for the 0.5° x 0.5° grid
surrounding the different stations during the **(a)** Belgica BG2014/14 and **(b)** GEOVIDE cruises. The dashed box
highlights the sampling period for both cruises (May 2014).

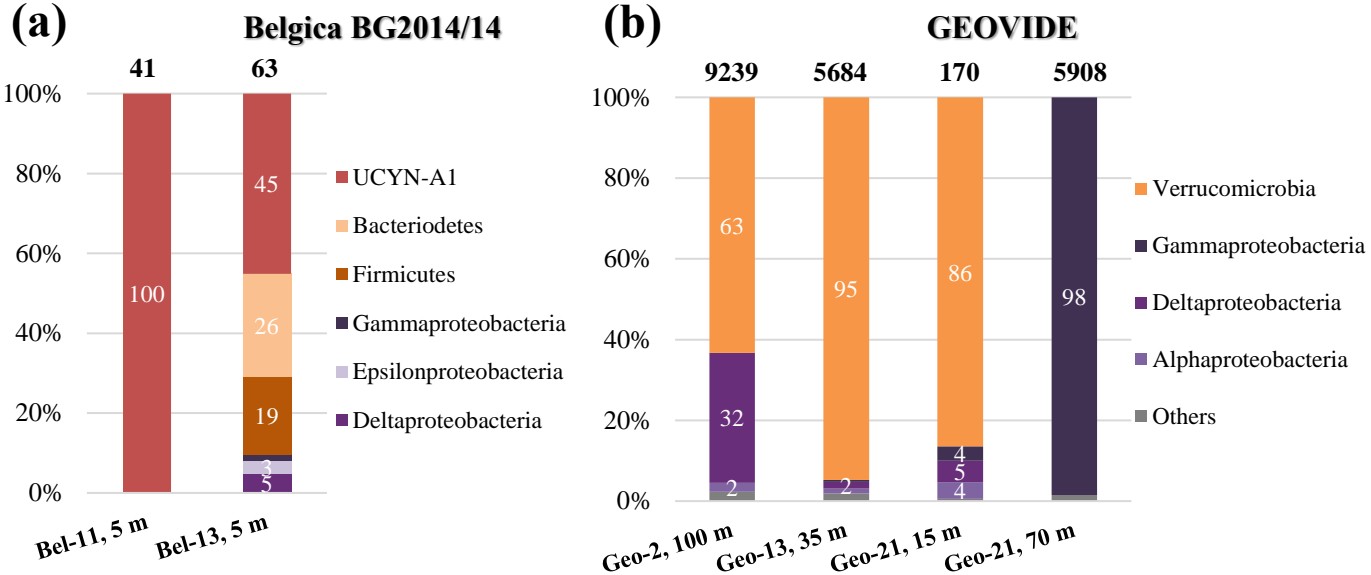

**Figure 5:** Diversity of *nifH* sequences during **(a)** the Belgica BG2014/14 cruise (successfully recovered only at stations Bel-11 and Bel-13, 5 m) and **(b)** the GEOVIDE cruise (stations Geo-2, 100 m; Geo-13, 35 m and Geo-21, 15 and 70 m. The total numbers of recovered sequences are indicated on top of the bars, and the exact percentage represented by each group is shown inside the bars.

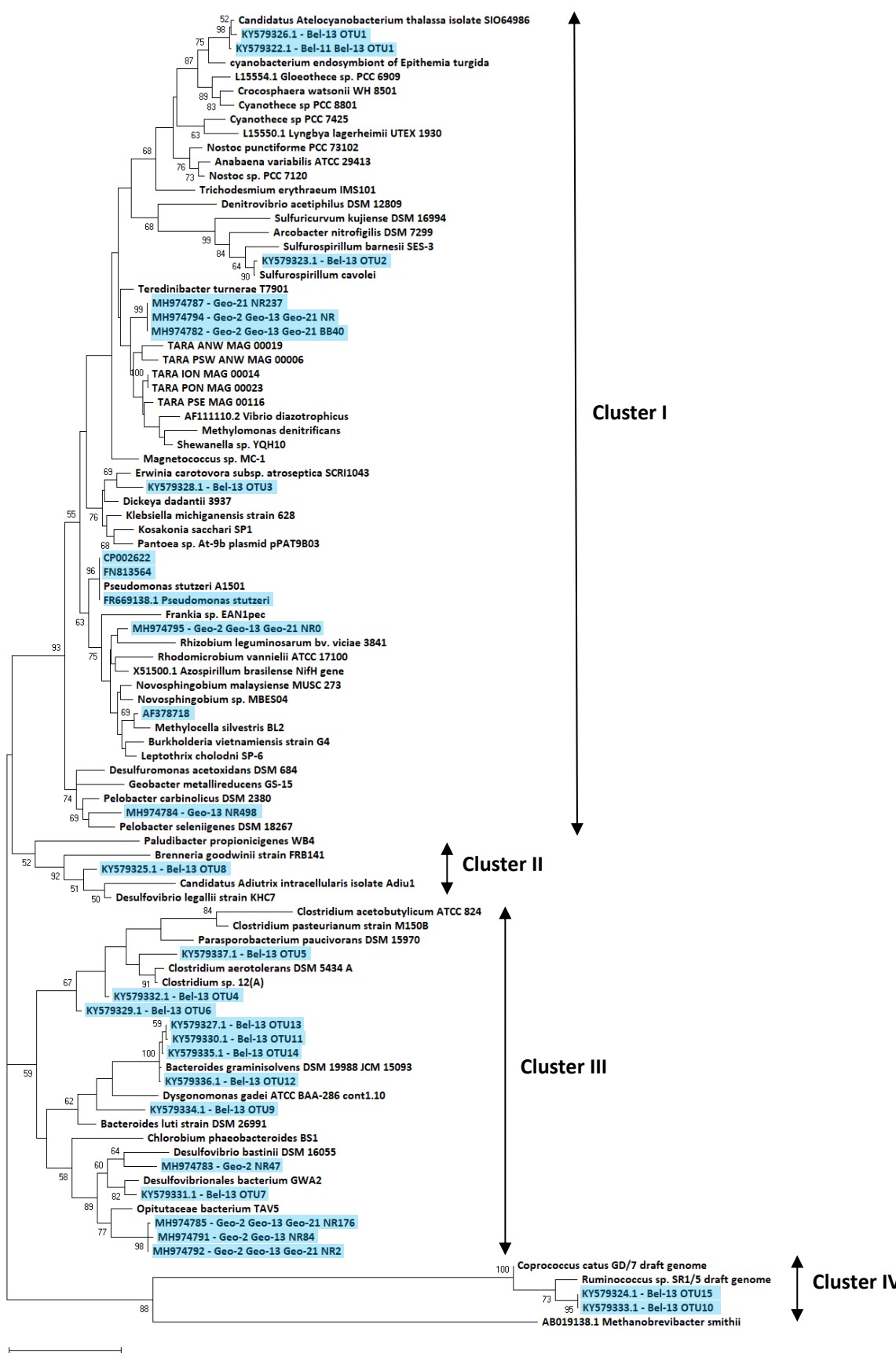

**Figure 6:** Phylogenetic tree of *nifH* predicted amino acid sequences generated using the Maximum Likelihood method of the Kimura 2-parameter model (Kimura, 1980) via the Molecular Evolutionary Genetics Analysis software

(MEGA 7.0) (Kumar et al., 2016). Initial tree(s) for the heuristic search were obtained automatically by applying
Neighbor-Join and BioNJ algorithms to a matrix of pairwise distances estimated using the Maximum Composite
Likelihood (MCL) approach, and then selecting the topology with superior log likelihood value. A discrete Gamma
distribution was used to model evolutionary rate differences among sites (5 categories (+G, parameter = 0.4038)). All
sequences recovered from DNA samples, including those previously identified and the newly recovered ones (with ≥
95% similarity at the nucleotide level with representative clones) are highlighted in blue. For the *nifH* sequences
recovered from the GEOVIDE cruise, only those contributing to the cumulative 98% of recovered sequences were
included in this tree. Bootstrap support values (≥ 50%) for 100 replications are shown at nodes. The scale bar
indicates the number of sequence substitutions per site. The archaean *Methanobrevibacter smithii* was used as an
outgroup. Accession numbers for published sequences used to construct the phylogenetic tree are given.


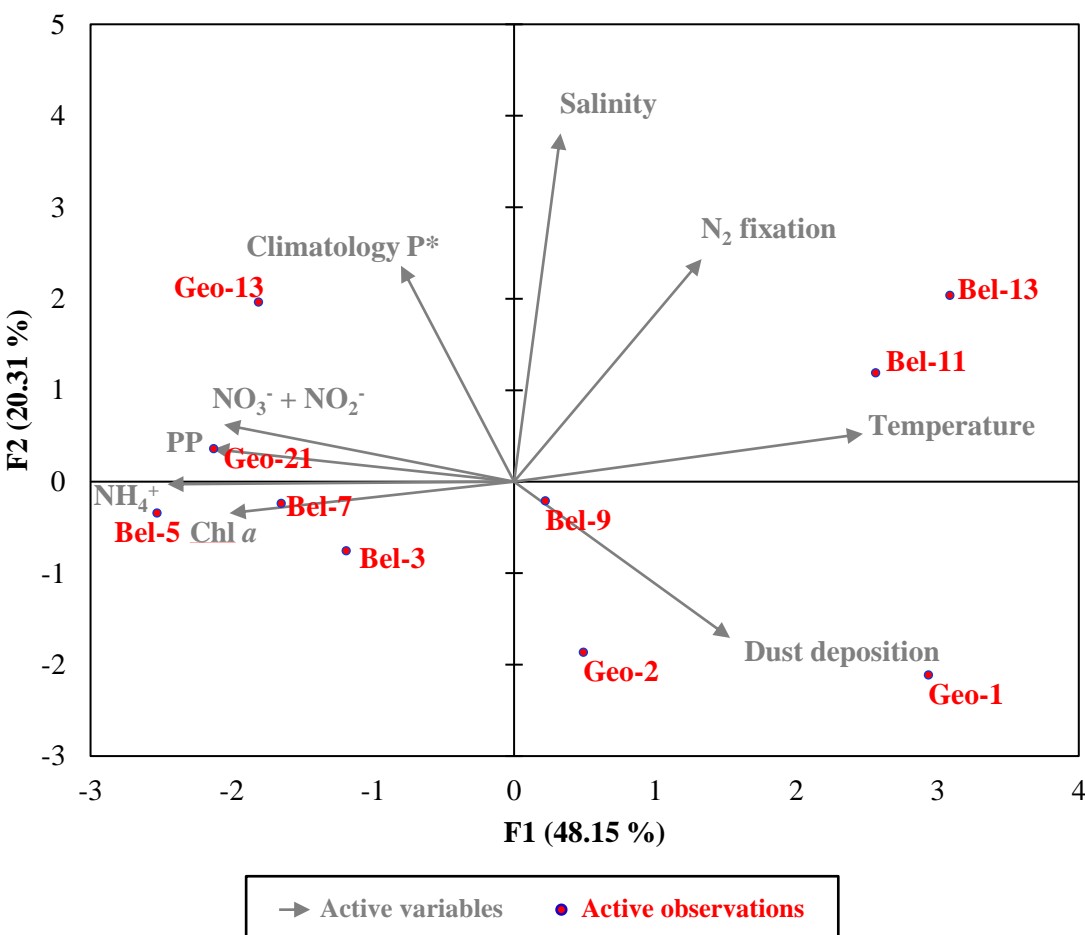

**Biplot (axes F1 and F2: 68.45 %)**

**Figure 7:** Euclidean distance biplot illustrating the axis loadings for the two main PCA components based on the Spearman rank correlation matrix shown in Table S3. Variables taken into account include depth-integrated rates of $N_2$ fixation and primary production (PP), average phosphate excess at 20 m depth surrounding each sampled site recovered from World Ocean Atlas 2013 climatology data between April and June from 1955 to 2012 (Garcia et al., 2013); satellite average dust deposition (dry + wet) derived during April 2014 (Giovanni online data system, NASA Goddard Earth Sciences Data and Information Services Center) and ambient variables (temperature, salinity, and nutrient data). Coloured dots in the biplot represent the projection of the different stations. Axis 1 has high negative loadings for PP, Chl $a$, $NH_4^+$ and $NO_3^- + NO_2^-$, and high positive loadings for temperature and $N_2$ fixation rates, with values of –0.812, –0.768, –0.936, –0.783, 0.942 and 0.506, respectively (see Table S5). Axis 2 has high positive loadings of 0.584, 0.943 and 0.602 for climatological P*, salinity and $N_2$ fixation rates, respectively. PCA analysis was run in XLSTAT 2017 (Addinsoft, Paris, France, 2017).