# Peer review of "Evidence of high N2 fixation rates in the temperate Northeast Atlantic"

_Biogeosciences, 2018_

## Editor Comment (EC1) · Z Jia (Editor) · 5 Aug 2018

This study was previously submitted to Frontiers in Microbiology (FiTM) and was rejected after critical reviews by two referees. Both reviewers are positive about the publication of this study in FiTM, although the comments are pretty critical, particularly reviewer#2. So the reason of rejection appears to be that the authors are not able to adequately address the concerns raised by the reviewers. Both reviewers are nice, despite of being somehow critical.

I have carefully read the comments and replies to the comments, and reviewed the ms myself. The manuscript could be published only after major revision suggested by both reviewers.

The key concerns are from reviewer#2, and I fully side with him/her. "Overall the text is too speculative- the discussion, although well written, is much too long with 10 pages, and especially the last 5 pages are full of speculation and partly wander too far from discussing actually measured data. It is impressive from how many different angles the authors try to shed light on their data, but when large parts of text end in hypothetical conclusions, assumptions, possibilities, it somehow makes the story implode back onto its core data, and it makes reading the long discussion a bit frustrating. I feel that the few interesting points could be summarized and put into necessary context in a much shorter way. I wonder whether the manuscript could be partly rewritten in order to focus more on the rates and less on the weak nifH data."

The major revisions should be made including:
(1) Stay focused on what you want to say. The authors need to focus on N2 fixation and productive water in this study, and get all relevant placed in the text first.
(2) The figures and tables need to be re-structured.

In summary, the major re-structuring is required as following.
(1) Table 1 might be useful, but table 2 can be placed in the supplementary text.

The key points includes:
(1) N2 fixation rate should be the figure 2, and primary production figure 3. Other figures are then arranged in an order to explain the N2 fixation and primary production
(2) In addition, Fig.7 and Fig.8 can be merged. Figure 4 and Figure 5 might be merged as well.
(3) Figure 6 may not be appropriate in this study because it will be published somewhere else.

Other key concerns for example are the following
(4) In the abstract, the table 1 was not mentioned which does not echo with the title "productive water"
(5) Line 24 delete the phrase of (38.8–46.5° N; 8.0–19.7° W) in May 2014
(6) L26-28. This statement is too descriptive
(7) L28. Delete the phrase In agreement with previous studies,

(8)  L35-38. Please provide solid evidence, instead of proposing something.

(9)  L39-40. These data appear to be placed in the supplementary materials. Therefore, it is no appropriate to be discussed in the abstract.

---

## Referee Comment (RC2) · Anonymous Referee #2 · 13 Aug 2018

Review of "Evidence of high N2 fixation rates in productive waters of the temperate Northeast Atlantic" by Fonseca-Batista et al.

The editor should understand that neither N fixation, N-fixing gene abundances, nor bioecology are within my realm of expertise. Despite that, I have sailed on many research cruises with N-fixation scientists, and even collected nifH samples for them on my own cruises. So I should be classified as a knowledgeable non-expert: I appreciate the research area but cannot analyze or critique details.

From that perspective, the paper is a fine contribution, calling attention to high N fixation rates and relatively high N-fixing gene copies in the temperate eastern Atlantic in the springtime following the spring bloom. This has not been observed before, partially due to methodological issues and partly due to the lack of observations in this season. Obviously it may change our view of how to assess global N fixation rates and how to model them.

I could only find a couple of small issues with the text (noted below), otherwise I believe it can be published with only minor revisions, hopefully with more guidance from a reviewer expert in this field.

(1) p. 1, line 31: For the sake of clarity, I recommend modifying the text to "At the sites where N2 fixation activity was the highest, we recovered sequences affiliated to UCYN-A1 (obligate symbiont of eukaryotic preymnesiophyte algae)."

(2) p. 5, line 161: the Ambar and Fiúza, 1994 paper is not in the list of references.

---

## Author Comment (AC1) · 24 Sep 2018

Biogeosciences Editor's comments

This study was previously submitted to Frontiers in Microbiology (FiTM) and was rejected after critical reviews by two referees. Both reviewers are positive about the publication of this study in FiTM, although the comments are pretty critical, particularly reviewer#2. So the reason of rejection appears to be that the authors are not able to adequately address the concerns raised by the reviewers. Both reviewers are nice, despite of being somehow critical. I have carefully read the comments and replies to the comments, and reviewed the ms myself. The manuscript could be published only

after major revision suggested by both reviewers.

We would like to thank the Editor for taking the time to review this manuscript himself. We have carefully considered the comments and have revised the manuscript accordingly. We addressed all the comments in a point-by-point manner.

The key concerns are from reviewer#2, and I fully side with him/her. "Overall the text is too speculative- the discussion, although well written, is much too long with 10 pages, and especially the last 5 pages are full of speculation and partly wander too far from discussing actually measured data. It is impressive from how many different angles the authors try to shed light on their data, but when large parts of text end in hypothetical conclusions, assumptions, possibilities, it somehow makes the story implode back onto its core data, and it makes reading the long discussion a bit frustrating. I feel that the few interesting points could be summarized and put into necessary context in a much shorter way. I wonder whether the manuscript could be partly rewritten in order to focus more on the rates and less on the weak nifH data".

We understand the Editor's concerns related to the comments made by the Reviewer #2 during processing by Frontiers in Marine Science (FiMS). However, we would like to point out that significant modifications had been made in the manuscript prior to the submission to Biogeosciences, and further adjustments were now implemented to focus the text on our data and less on hypothetical environmental drivers: (1) the discussion section is now 5 pages (compared to the initial 10 pages mentioned above). The most criticized section related to the environmental drivers of N2 fixation now only has 2 pages (instead of 5). (2) the main hypotheses put forward to explain the distribution and magnitude of N2 fixation rates observed in this study were better summarized, made more concise and backed up with additional data from the GEOVIDE cruise that were not included in the last version submitted to FiMS. These supporting data include particulate organic carbon concentration in the surface waters and export efficiency (reflecting the potential for shallow recycling) (Lemaitre et al., under review in this same BG special issue for the GEOVIDE project), dissolved iron concentrations (Tonnard

et al., under review in the GEOVIDE special issue), and nifH data on the diazotroph community in the Iberian Basin and West European Basin.

Please note that the nifH sequences recovered from the GEOVIDE sampling have recently been submitted to the GenBank database, and we will soon have the accession numbers for them. These will be added to the manuscript (methods section 2.3) and the phylogenetic tree in Supplementary Figure S1, before publication, if the manuscript is accepted.

The major revisions should be made including: (1) Stay focused on what you want to say. The authors need to focus on N2 fixation and productive water in this study, and get all relevant placed in the text first. (2) The figures and tables need to be restructured. In summary, the major re-structuring is required as following. (1) Table 1 might be useful, but table 2 can be placed in the supplementary text.

A major restructuring of the figures and tables was implemented as suggested by the Editor: - both pair of Figures (Fig. 4-5 and Fig. 7-8) have now been merged. - Fig. 6 was removed from the article, the phytoplankton pigments data is now only cited as part of another manuscript in preparation, as follows: Tonnard et al., in preparation. - Table 2 was transferred to supplementary material.

Based on the BG Editor's latest comments, texts were adapted, particularly in the abstract, introduction, methods and results (to add DNA analysis and results from GEOVIDE cruise) and finally in the discussion to focus more on N2 fixation rates, their significance at the regional and basin scales, and their contribution to primary production.

The key points includes: (1) N2 fixation rate should be the figure 2, and primary production figure 3. Other figures are then arranged in an order to explain the N2 fixation and primary production (2) In addition, Fig.7 and Fig.8 can be merged. Figure 4 and Figure 5 might be merged as well. (3) Figure 6 may not be appropriate in this study because it will be published somewhere else.

Figures and Tables were re-arranged as suggested (see above). Concerning the order of figure presentation, we preferred discussing the sections on $\theta$/S diagrams and nutrients and chlorophyll prior to the rates measurements. We believe that introducing the physico-chemical features of the studied region not only allows one to easily grasp the environmental context of the study, but also to highlight specific regions with characteristic traits of interest which will not need to be detailed when discussing N2 fixation rates and Primary Production.

Other key concerns for example are the following (4) In the abstract, the table 1 was not mentioned which does not echo with the title "productive water"

We agree, the abstract was adapted to focus more on N2 fixation rates, their contribution to primary production, but also to reduce the attention brought to specific hypothetical environmental drivers.

(5) Line 24 delete the phrase of (38.8–46.5° N; 8.0–19.7° W) in May 2014

The latitudinal and longitudinal range of the studied region was deleted, we just kept "May 2014" to indicate the time of sampling which is relevant based on the importance of seasonality in this work, a condition now better highlighted in the abstract (line 23)

(6) L26-28. This statement is too descriptive

The sentence was modified, to bring the reader's attention to the facts that (1) such high N2 fixation rates have not yet been observed earlier along the whole eastern Atlantic boundary, (2) elevated rates like these have only been reported at the western Atlantic boundary in very specific nutrient-rich environments such as: coastal, shelf and mesohaline waters. The sentence now reads as follows (lines 29-31):

"In the Atlantic Ocean, N2 fixation rates exceeding 1000 $\mu$mol N m-2 d-1 have previously only been reported in the temperate and tropical western North Atlantic waters having coastal, shelf or mesohaline characteristics, as opposed to the mostly open ocean conditions studied here."

(7) L28. Delete the phrase In agreement with previous studies,

The sentence was deleted and the text was adapted as follows (lines 31-35 and 37-39):

"At the two sites where N2 fixation activity was the highest; nifH sequences assigned to the prymnesiophyte-symbiont Candidatus Atelocyanobacterium thalassa (UCYN-A) dominated the nifH sequence pool recovered from DNA samples, while the remaining sequences, as for all the ones recovered from the other sites, belonged exclusively to non-cyanobacterial phylotypes." ... "Earlier studies in the Iberian region were conducted largely outside the bloom period, unlike the present work which was carried out in spring, yet in all cases the assessment of nifH gene diversity, suggests a predominance of UCYN-A and non-cyanobacterial diazotrophs."

(8) L35-38. Please provide solid evidence, instead of proposing something.

The hypotheses are now introduced along with supporting information presented in the corresponding discussion section, regarding the abundance of particulate organic carbon in surface waters (based on Lemaitre et al., under review in the same GEOVIDE BG Special Issue), and on the in situ excess phosphorus data. The text was adapted as follows (lines 35-43):

"Previous studies in the Iberian Basin have systematically reported lower N2 fixation rates (from < 0.1 to 140 $\mu$mol N m-2 d-1), as compared to those found in the present study, and this regardless of whether the bubble-addition method or the dissolution method were applied. Earlier studies in the Iberian region were conducted largely outside the bloom period, unlike the present work which was carried out in spring, yet in all cases the assessment of nifH gene diversity, suggests a predominance of UCYN-A and non-cyanobacterial diazotrophs. We support that the unexpectedly high N2 fixation activities recorded at the time of our study were promoted by the availability of phytoplankton-derived organic matter produced during the spring bloom, as evidenced by the significant surface particulate organic carbon concentrations, and by the presence of excess phosphorus signature in surface waters, particularly at the sites with

extreme activities."

(9) L39-40. These data appear to be placed in the supplementary materials. Therefore, it is no appropriate to be discussed in the abstract.

We agree, that sentence was deleted. 

Please also note the supplement to this comment:
https://www.biogeosciences-discuss.net/bg-2018-220/bg-2018-220-AC1-supplement.pdf

[Figure]

**Figure 4**

**Fig. 1.** Figure 4: Spatial distribution of depth-integrated primary production

[Figure]

[Figure]

Figure 5

**Fig. 2.** Figure 5: Spatial distribution of depth-integrated N2 fixation

**Supplement:**

**Evidence of high N₂ fixation rates in productive waters of the temperate Northeast Atlantic**

[revised manuscript text omitted]

**2.1 Environmental conditions**

Temperature, salinity and photosynthetically active radiation (PAR) profiles were determined using a conductivity-temperature-depth sensor (SBE 09 and SBE 911+, during the BG2014/14 and GEOVIDE cruises, respectively) fitted on a rosette equipped with either 12 or 24 Niskin bottles to sample seawater for biogeochemical measurements. Water column concentrations of ammonium ($NH_4^+$) during both cruises were measured on board as well as nitrate + nitrite ($NO_3^- + NO_2^-$) concentrations during the GEOVIDE expedition. During the BG2014/14 cruise, samples dedicated for $NO_3^- + NO_2^-$ and phosphate ($PO_4^{3-}$) measurements were filtered (0.2 µm) and stored at –20°C until analysis at the home-based laboratory. $PO_4^{3-}$ data are not yet available for the GEOVIDE cruise.

Nutrient concentrations were determined using the conventional fluorometric (for $NH_4^+$) (Holmes et al., 1999) and colorimetric methods (for the other nutrients) (Grasshoff et al., 1983) with detection limits (DL) of 64 nmol $L^{-1}$ ($NH_4^+$), 90 nmol $L^{-1}$ ($NO_3^- + NO_2^-$) and 60 nmol $L^{-1}$ ($PO_4^{3-}$). For the BG2014/14 cruise, chlorophyll *a* (Chl *a*) concentrations were determined according to Yentsch and Menzel (1963), by filtering 250 mL of seawater sample onto Whatman GF/F glass microfiber filters (0.7 µm nominal pore size), followed by pigment extraction in 90% acetone, centrifugation and fluorescence measurement using a Shimadzu RF-150 fluorometer.

**2.2 $^{15}N_2$ fixation and $^{13}C\text{-}HCO_3^-$ uptake rates**

$N_2$ fixation and primary production (PP) were determined simultaneously in duplicate using the $^{15}N\text{-}N_2$ dissolution method (Großkopf et al., 2012) and $^{13}C\text{-}NaHCO_3$ tracer addition (Hama et al., 1983) techniques. Seawater samples

were collected in 4.5 L acid-cleaned polycarbonate (PC) bottles from a minimum of four depths (six at stations Geo-1, Geo-13 and Geo-21) equivalent to 54%, 13%, 3% and 0.2% of surface PAR (plus 25% and 1% PAR for Geo-1, Geo-13 and Geo-21). Details concerning the applied $^{15}N_2$ dissolution method can be found in Fonseca-Batista et al.

[revised manuscript text omitted]

The predominant upper layer water mass in this basin is the Eastern North Atlantic Central Water (ENACW), a winter mode water which consists of two components, according to Fiúza (1984) (see θ/S diagrams, Fig. 2): (i) the

190 lighter, relatively warm and salty ENACWst formed in the subtropical Azores Front region (~35° N) when Azores Mode Water is subducted as a result of strong evaporation and winter cooling; and (ii) the colder and less saline ENACWsp, underlying the ENACWst, and formed in the subpolar eastern North Atlantic (north of 43° N) through winter cooling and deep convection (McCartney and Talley, 1982). The spatial distribution of these Central Waters allowed categorizing the sampling sites in 2 groups: (i) ENACWsp stations north of 43° N (Bel-3, Bel-5, Bel-7, and

195 Geo-21) only affected by the ENACWsp (Fig. 2a, b) and (ii) ENACWst stations, south of 43° N, characterized by the upper layer being influenced by ENACWst and the subsurface layer by ENACWsp (Fig. 2a, b). Most of these ENACWst stations are open ocean sites (Bel-9, Bel-11, Bel-13, and Geo-13) while two are influenced by their proximity to the shelf (Geo-1 and Geo-2) (Tonnard et al., 2018).

Surface waters of all the ENACWst stations showed a relatively strong stratification resulting from the progressive

200 spring heating, with sea surface temperature (SST) ranging from 15.3 (Geo-13) to 17.2°C (Bel-13). Nutrients were depleted at the surface (NO$_3^-$ + NO$_2^-$ < 0.09 µM in the upper 20 m; Fig. 3c, f) and surface Chl *a* concentrations were low (< 0.25 µg L$^{-1}$; Fig. 3a, d) but showed a subsurface maximum (between 0.5 and 0.75 µg L$^{-1}$ at approximately 50

m), a common feature for oligotrophic open ocean waters. Amongst the ENACWst stations, station Geo-13 had a slightly higher nutrient content ($NO_3^-$ + $NO_2^-$ = 0.7 µM in the lower mixed layer depth, MLD) and higher Chl $a$ (> 0.5 µg L$^{-1}$ in the upper 35 m).

Surface waters at ENACWsp stations were less stratified (SST between 14.0 and 14.5°C), were nutrient replete (surface $NO_3^-$ + $NO_2^-$ ranging from 0.3 to 0.8 µM) and had a higher phytoplankton biomass (Chl $a$ between 0.7 to 1.2 µg L$^{-1}$ in the upper 30 m except for station Bel-5). Highest Chl $a$ values were observed at station Bel-7 (44.6° N, 9.3° W), which appeared to be located within an anticyclonic mesoscale eddy, as evidenced by the downwelling structure detected in the Chl $a$ and $NO_3^-$ + $NO_2^-$ profiles (Fig. 3a, c) at this location (as well as T and S sections, data not shown).

**3.2 Primary production and pigment distribution**

Volumetric rates of carbon uptake ranged from 7 to 3500 µmol C m$^{-3}$ d$^{-1}$ (see Supporting Information Table S1) and euphotic layer integrated rates varied from 32 to 137 mmol C m$^{-2}$ d$^{-1}$ (Fig. 4a, b, and Supporting Information Table S2).

PP was relatively homogenous in the Bay of Biscay (stations Bel-3, Bel-5 and Bel-7) and along the Iberian Margin (Bel-9, Bel-11, Bel-13 and Geo-1) with average rates ranging from 33 to 43 mmol C m$^{-2}$ d$^{-1}$, except at station Bel-7 where it was slightly higher (52 mmol C m$^{-2}$ d$^{-1}$; Fig. 4a, b, and Supporting Information Table S2), likely due to the presence of an anticyclonic mesoscale structure at this location. PP increased westwards away from the Iberian Peninsula, reaching highest values at stations Geo-13 and Geo-21 (79 and 135 mmol C m$^{-2}$ d$^{-1}$, respectively; Fig. 4b) as well as closer to the shelf (reaching 59 mmol C m$^{-2}$ d$^{-1}$ at Geo-2). These results are in the range of past measurements for the same period of the year, ranging from 19 to 103 mmol C m$^{-2}$ d$^{-1}$ (Marañón et al., 2000; Fernández et al., 2005; Poulton et al., 2006; Fonseca-Batista et al., 2017). Our observations also coincide with the area-averaged Chl $a$ time series obtained from satellite data (Giovanni online data system; Fig. 4c, d) which reveal that post bloom conditions prevailed at most sites (Bel-3 to Bel-13 and Geo-1 to Geo-13) while the bloom was still ongoing at station Geo-21 at the time of our study. Higher PP rates appear to coincide with the increase, offshore and towards the shelf, of the relative abundance of diatoms, based on fucoxanthin pigment concentrations (Tonnard et al., in preparation). At the GEOVIDE sites exhibiting lowest fixed-nitrogen concentrations, Geo-1 and Geo-13, prymnesiophytes represented 30–40% of the phytoplankton community, compared to 20–35% at stations Geo-21 and Geo-2 (based on the presence of 19'-hexanoyloxyfucoxanthin pigment). Such relative abundances are in agreement with the global abundance of prymnesiophytes ($32 \pm 5\%$) proposed by Swan et al. (2016).

**3.3 N$_2$ fixation and dominant diazotrophs at the sampling sites**

Volumetric N$_2$ fixation rates were above the detection limit at 8 of the 10 stations sampled in this study (excluding Bel-3 and Bel-5 where rates were below the detection limit) and ranged from 0.7 to 65.4 nmol N L$^{-1}$ d$^{-1}$ (see Supporting Information Table S1), with areal rates ranging between 81 and 1533 µmol N m$^{-2}$ d$^{-1}$ (Fig. 5a, b, and Supporting Information Table S2).

We observed intense N$_2$ fixation activities at the two sites (Bel-11 and Bel-13) most affected by ENACW waters of subtropical origin (Fig. 2). At stations Bel-11 and Bel-13, volumetric rates of N$_2$ fixation ranged from 2.4 to 65.4 nmol N L$^{-1}$ d$^{-1}$, with highest rates found at surface level (65.4 and 45.0 nmol N L$^{-1}$ d$^{-1}$, respectively), while areal rates averaged 1533 and 1355 µmol N m$^{-2}$ d$^{-1}$, respectively. N$_2$ fixation was also relatively high at the most productive sites

Bel-7 and Geo-21 with volumetric rates ranging from 1.0 to 8.2 nmol N $L^{-1}$ $d^{-1}$ and areal rates averaging 128 and 279 µmol N $m^{-2}$ $d^{-1}$, respectively. In contrast to the Belgica sites, $
[revised manuscript text omitted]

Luo, Y.-W., Doney, S. C., Anderson, L. A., Benavides, M., Berman-Frank, I., Bode, A., … Zehr, J. P. (2012). Database of diazotrophs in global ocean: abundance, biomass and nitrogen fixation rates. Earth System Science Data, 4(1), 47–73. https://doi.org/10.5194/essd-4-47-2012

Luo, Y.-W., Lima, I. D., Karl, D. M., Deutsch, C. A., & Doney, S. C. (2014). Data-based assessment of environmental controls on global marine nitrogen fixation. Biogeosciences, 11(3), 691–708. https://doi.org/10.5194/bg-11-691-2014

Man-Aharonovich, D., Kress, N., Zeev, E. B., Berman-Frank, I., & Béjà, O. (2007). Molecular ecology of nifH genes and transcripts in the eastern Mediterranean Sea. Environmental Microbiology, 9(9), 2354–2363. https://doi.org/10.1111/j.1462-2920.2007.01353.x

Marañón, E., Holligan, P. M., Varela, M., Mouriño, B., & Bale, A. J. (2000). Basin-scale variability of phytoplankton biomass, production and growth in the Atlantic Ocean. Deep Sea Research Part I: Oceanographic Research Papers. https://doi.org/10.1016/S0967-0637(99)00087-4

Martínez-Pérez, C., Mohr, W., Löscher, C. R., Dekaezemacker, J., Littmann, S., Yilmaz, P., … Kuypers, M. M. M. (2016). The small unicellular diazotrophic symbiont, UCYN-A, is a key player in the marine nitrogen cycle. Nature Microbiology, 1(September), 1–7. https://doi.org/10.1038/nmicrobiol.2016.163

McCartney, M. S., & Talley, L. D. (1982). The Subpolar Mode Water of the North Atlantic Ocean. Journal of Physical Oceanography. https://doi.org/10.1175/1520-0485(1982)012<1169:TSMWOT>2.0.CO;2

695 Mills, M. M., Ridame, C., Davey, M., La Roche, J., & Geider, R. J. (2004). Iron and phosphorus co-limit nitrogen fixation in the eastern tropical North Atlantic. Nature, 429(May), 292–294. https://doi.org/10.1038/nature02550

Mohr, W., Großkopf, T., Wallace, D. W. R., & LaRoche, J. (2010). Methodological underestimation of oceanic nitrogen fixation rates. PLoS ONE, 5(9), 1–7. https://doi.org/10.1371/journal.pone.0012583

Montoya, J. P., Voss, M., & Capone, D. G. (2007). Spatial variation in N2-fixation rate and diazotroph activity in the

700 Tropical Atlantic. Biogeosciences, 4(3), 369–376. https://doi.org/10.5194/bg-4-369-2007

Montoya, J. P., Voss, M., Kahler, P., & Capone, D. G. (1996). A Simple , High-Precision , High-Sensitivity Tracer Assay for N2 Fixation. Applied and Environmental Microbiology, 62(3), 986–993.

Moore, C. M., Mills, M. M., Achterberg, E. P., Geider, R. J., LaRoche, J., Lucas, M. I., … Woodward, E. M. S. (2009). Large-scale distribution of Atlantic nitrogen fixation controlled by iron availability. Nature Geoscience,

705 2(12), 867–871. https://doi.org/10.1038/ngeo667

Moreira-Coello, V., Mouriño-Carballido, B., Marañón, E., Fernández-Carrera, A., Bode, A., & Varela, M. M. (2017). Biological N2 Fixation in the Upwelling Region off NW Iberia: Magnitude, Relevance, and Players. Frontiers in Marine Science, 4(September). https://doi.org/10.3389/fmars.2017.00303

Mulholland, M. R., Bernhardt, P. W., Blanco-Garcia, J. L., Mannino, A., Hyde, K., Mondragon, E., … Zehr, J. P.

710 (2012). Rates of dinitrogen fixation and the abundance of diazotrophs in North American coastal waters between Cape Hatteras and Georges Bank. Limnology and Oceanography, 57(4), 1067–1083. https://doi.org/10.4319/lo.2012.57.4.1067

Needoba, J. A., Foster, R. A., Sakamoto, C., Zehr, J. P., & Johnson, K. S. (2007). Nitrogen fixation by unicellular diazotrophic cyanobacteria in the temperate oligotrophic NorthA.Pacific Ocean. Limnology and Oceanography,

715 52(4), 1317–1327. https://doi.org/10.4319/lo.2007.52.4.1317

Nei, M. (1987). Molecular Evolutionary Genetics. Tempe AZ Arizona State University (Vol. 17). Columbia University Press, New York, USA.

Ohlendieck, U., Stuhr, A., & Siegmund, H. (2000). Nitrogen fixation by diazotrophic cyanobacteria in the Baltic Sea and transfer of the newly fixed nitrogen to picoplankton organisms. Journal of Marine Systems, 25(3–4), 213–219.

720 https://doi.org/10.1016/S0924-7963(00)00016-6

Poulton, A. J., Holligan, P. M., Hickman, A., Kim, Y. N., Adey, T. R., Stinchcombe, M. C., … Woodward, E. M. S. (2006). Phytoplankton carbon fixation, chlorophyll-biomass and diagnostic pigments in the Atlantic Ocean. Deep-Sea Research Part II: Topical Studies in Oceanography, 53(14–16), 1593–1610. https://doi.org/10.1016/j.dsr2.2006.05.007

725 Rahav, E., Bar-Zeev, E., Ohayon, S., Elifantz, H., Belkin, N., Herut, B., … Berman-Frank, I. (2013). Dinitrogen fixation in aphotic oxygenated marine environments. Frontiers in Microbiology, 4(AUG), 1–11. https://doi.org/10.3389/fmicb.2013.00227

Rahav, E., Giannetto, M. J., & Bar-Zeev, E. (2016). Contribution of mono and polysaccharides to heterotrophic N2 fixation at the eastern Mediterranean coastline. Scientific Reports, 6(May), 1–11. https://doi.org/10.1038/srep27858

730 Ratten, J.-M. (2017). The diversity, distribution and potential metabolism of non-cyanobacterial diazotrophs in the North Atlantic Ocean. Dalhousie University.

Ratten, J. M., LaRoche, J., Desai, D. K., Shelley, R. U., Landing, W. M., Boyle, E., … Langlois, R. J. (2015). Sources of iron and phosphate affect the distribution of diazotrophs in the North Atlantic. Deep-Sea Research Part II: Topical Studies in Oceanography, 116, 332–341. https://doi.org/10.1016/j.dsr2.2014.11.012

735    Raven, J. A. (1988). The iron and molybdenum use efficiencies of plant growth with different energy, carbon and nitrogen sources. New Phytologist, 109, 279–287. https://doi.org/10.1111/j.1469-8137.1988.tb04196.x

Rees, A., Gilbert, J., & Kelly-Gerreyn, B. (2009). Nitrogen fixation in the western English Channel (NE Atlantic Ocean). Marine Ecology Progress Series, 374(1979), 7–12. https://doi.org/10.3354/meps07771

Rijkenberg, M. J. A., Langlois, R. J., Mills, M. M., Patey, M. D., Hill, P. G., Nielsdóttir, M. C., … Achterberg, E. P.
740    (2011). Environmental forcing of nitrogen fixation in the Eastern Tropical and Sub-Tropical North Atlantic Ocean. PLoS ONE, 6(12). https://doi.org/10.1371/journal.pone.0028989

Riou, V., Fonseca-Batista, D., Roukaerts, A., Biegala, I. C., Prakya, S. R., Magalhães Loureiro, C., … Dehairs, F. (2016). Importance of N2-Fixation on the Productivity at the North-Western Azores Current/Front System, and the Abundance of Diazotrophic Unicellular Cyanobacteria. Plos One, 11(3), e0150827.
745    https://doi.org/10.1371/journal.pone.0150827

Sarthou, G., Baker, A. R., Blain, S., Achterberg, E. P., Boye, M., Bowie, A. R., … Worsfold, P. J. (2003). Atmospheric iron deposition and sea-surface dissolved iron concentrations in the eastern Atlantic Ocean. Deep-Sea Research Part I: Oceanographic Research Papers, 50(10–11), 1339–1352. https://doi.org/10.1016/S0967-0637(03)00126-2

750    Scavotto, R. E., Dziallas, C., Bentzon-Tilia, M., Riemann, L., & Moisander, P. H. (2015). Nitrogen-fixing bacteria associated with copepods in coastal waters of the North Atlantic Ocean. Environmental Microbiology, 17(10), 3754–3765. https://doi.org/10.1111/1462-2920.12777

Shelley, R. U., Roca-Martí, M., Castrillejo, M., Sanial, V., Masqué, P., Landing, W. M., … Sarthou, G. (2017). Quantification of trace element atmospheric deposition fluxes to the Atlantic Ocean (> 40°N; GEOVIDE,
755    GEOTRACES GA01) during spring 2014. Deep-Sea Research Part I, 119(November 2016), 34–49. https://doi.org/10.1016/j.dsr.2016.11.010

Shiozaki, T., Ijichi, M., Kodama, T., Takeda, S., Furuya, K., Ijichi, M., … Furuya, K. (2014). Heterotrophic bacteria as major nitrogen fixers in the euphotic zone of the Indian Ocean. Global Biogeochemical Cycles, 28, 1096–1110. https://doi.org/10.1002/2014GB004886.Received

760    Shiozaki, T., Nagata, T., Ijichi, M., & Furuya, K. (2015). Nitrogen fixation and the diazotroph community in the temperate coastal region of the northwestern North Pacific. Biogeosciences, 12(15), 4751–4764. https://doi.org/10.5194/bg-12-4751-2015

Snow, J. T., Schlosser, C., Woodward, E. M. S., Mills, M. M., Achterberg, E. P., Mahaffey, C., … Moore, C. M. (2015). Environmental controls on the biogeography of diazotrophy and Trichodesmium in the Atlantic Ocean.
765    Global Biogeochemical Cycles, 29, 865–884. https://doi.org/10.1002/2013GB004679.Received

Subramaniam, A., Mahaffey, C., Johns, W., & Mahowald, N. (2013). Equatorial upwelling enhances nitrogen fixation in the Atlantic Ocean. Geophysical Research Letters, 40(9), 1766–1771. https://doi.org/10.1002/grl.50250

Subramaniam, A., Yager, P. L., Carpenter, E. J., Mahaffey, C., Björkman, K., Cooley, S., … Capone, D. G. (2008). Amazon River enhances diazotrophy and carbon sequestration in the tropical North Atlantic Ocean. Global
770    Biogeochemical Cycles, 105, 10460–10465. https://doi.org/10.1029/2006GB002751

Swan, C. M., Vogt, M., Gruber, N., & Laufkoetter, C. (2016). A global seasonal surface ocean climatology of phytoplankton types based on CHEMTAX analysis of HPLC pigments. Deep-Sea Research Part I: Oceanographic Research Papers, 109, 137–156. https://doi.org/10.1016/j.dsr.2015.12.002

Thompson, A. W., Foster, R. A., Krupke, A., Carter, B. J., Musat, N., Vaulot, D., … Zehr, J. P. (2012). Unicellular
775    Cyanobacterium Symbiotic with a Single-Celled Eukaryotic Alga. Science, 337(September), 1546–1550.

Thuróczy, C.-E., Gerringa, L. J. A., Klunder, M. B., Middag, R., Laan, P., Timmermans, K. R., & de Baar, H. J. W. (2010). Speciation of Fe in the Eastern North Atlantic Ocean. Deep Sea Research Part I: Oceanographic Research Papers, 57(11), 1444–1453. https://doi.org/10.1016/j.dsr.2010.08.004

Tonnard M., Donval A., Lampert L., Treguer P., Claustre H., Dimier C., Ras J., Sauzède R., Foliot L., Bowie A. R., van der Merwe P., Planquette H., and Sarthou G. Phytoplankton assemblages in the North Atlantic Ocean and in the Labrador Sea (GEOTRACES, GA01) determined by CHEMTAX analysis: Assessment of the community structure, succession and potential limitation. In preparation

Tonnard, M., Planquette, H., Bowie, A. R., van der Merwe, P., Gallinari, M., Deprez de Gesincourt, F., … Sarthou, G. (2018). Dissolved iron in the North Atlantic Ocean and Labrador Sea along the GEOVIDE section (GEOTRACES section GA01). Submitted to this Biogeosciences Special Issue, https://doi.org/10.5194/bg-2018-147

Vidal, M., Duarte, C. M., & Agustí, S. (1999). Dissolved organic nitrogen and phosphorus pools and fluxes in the central Atlantic Ocean. Limnology and Oceanography, 44(1), 106–115.

Vidal, M., Duarte, C. M., Agustí, S., Gasol, J. M., & Vaqué, D. (2003). Alkaline phosphatase activities in the central Atlantic Ocean indicate large areas with phosphorus deficiency. Marine Ecology Progress Series, 262, 43–53. https://doi.org/10.3354/meps262043

Villa-Alfageme, M., de Soto, F. C., Ceballos, E., Giering, S. L. C., Le Moigne, F. A. C., Henson, S., … Sanders, R. J. (2016). Geographical, seasonal, and depth variation in sinking particle speeds in the North Atlantic. Geophysical Research Letters, 43, 8609–8616. https://doi.org/10.1002/2016GL069233.Received

Voss, M., Croot, P., Lochte, K., Mills, M., & Peeken, I. (2004). Patterns of nitrogen fixation along 10°N in the tropical Atlantic. Geophysical Research Letters, 31(23), 1–4. https://doi.org/10.1029/2004GL020127

Dyhrman, S. T., Chappell, P. D., Haley, S. T., Moffett, J. W., Orchard, E. D., Waterbury, J. B., & Webb, E. A. (2006). Phosphonate utilization by the globally important marine diazotroph Trichodesmium. Nature, 439(7072), 68–71. https://doi.org/10.1038/nature04203

Dyhrman, S. T., & Haley, S. T. (2006). Phosphorus scavenging in the unicellular marine diazotroph Crocosphaera watsonii phosphorus scavenging in the unicellular marine diazotroph Crocosphaera watsonii. Applied and Environmental Microbiology, 72(2), 1452–1458. https://doi.org/10.1128/AEM.72.2.1452

Khadem, A. F., Pol, A., Jetten, M. S. M., & Op Den Camp, H. J. M. (2010). Nitrogen fixation by the verrucomicrobial methanotroph "Methylacidiphilum fumariolicum" SolV. Microbiology, 156(4), 1052–1059. https://doi.org/10.1099/mic.0.036061-0

Lemaitre, N., Planchon, F., Planquette, H., Dehairs, F., Fonseca-Batista, D., Roukaerts, A., … Sarthou, G. (2018). High variability of export fluxes along the North Atlantic GEOTRACES section GA01: Particulate organic carbon export deduced from the 234Th method. Biogeosciences Discussions, (April), 1–38. https://doi.org/10.5194/bg-2018-190

Li, X., Fonseca-Batista, D., Roevros, N., Dehairs, F., & Chou, L. (2018). Environmental and nutrient controls of marine nitrogen fixation. Progress in Oceanography, 167(August), 125–137. https://doi.org/10.1016/j.pocean.2018.08.001

Luo, H., Benner, R., Long, R. a, & Hu, J. (2009). Subcellular localization of marine bacterial alkaline phosphatases. Proceedings of the National Academy of Sciences of the United States of America, 106(50), 21219–21223. https://doi.org/10.1073/pnas.0907586106

Mulholland, M. R., Bernhardt, P. W., Blanco-Garcia, J. L., Mannino, a., Hyde, K., Mondragon, E., … Zehr, J. P. (2012). Rates of dinitrogen fixation and the abundance of diazotrophs in North American coastal waters between

Cape Hatteras and Georges Bank. Limnology and Oceanography, 57(4), 1067–1083. https://doi.org/10.4319/lo.2012.57.4.1067

820    Tonnard, M., Planquette, H., Bowie, A. R., van der Merwe, P., Gallinari, M., Deprez de Gesincourt, F., … Sarthou, G. (2018). Dissolved iron distribution in the North Atlantic Ocean and Labrador Sea along the GEOVIDE section (GEOTRACES section GA01). Biogeosciences Discussions, (April).

Wertz, J. T., Kim, E., Breznak, J. A., Schmidt, T. M., & Rodrigues, J. L. M. (2012). Genomic and physiological characterization of the Verrucomicrobia isolate Diplosphaera colitermitum gen. nov., sp. nov., reveals

825    microaerophily and nitrogen fixation genes. Applied and Environmental Microbiology, 78(5), 1544–1555. https://doi.org/10.1128/AEM.06466-11

Xu, Y., Wahlund, T. M., Feng, L., Shaked, Y., & Morel, F. M. M. (2006). A novel alkaline phosphatase in the coccolithophore Emiliania huxleyi (Prymnesiophyceae) and its regulation by phosphorus. Journal of Phycology, 42(4), 835–844. https://doi.org/10.1111/j.1529-8817.2006.00243.x

830    Yentsch, C. S., & Menzel, D. W. (1963). A method for the determination of phytoplankton chlorophyll and phaeophytin by fluorescence. Deep Sea Research and Oceanographic Abstracts, 10(3), 221–231. https://doi.org/10.1016/0011-7471(63)90358-9

Zani, S., Mellon, M. T., Collier, J. L., & Zehr, J. P. (2000). Expression of nifH genes in natural microbial assemblages in Lake George, New York, detected by reverse transcriptase PCR. Applied and Environmental

835    Microbiology, 66(7), 3119–3124. https://doi.org/10.1128/AEM.66.7.3119-3124.2000

**Tables**

 **Table 1:** Relative contribution (%) of $N_2$ fixation to Primary Production (PP).

| Province | Station | Latitude (° N) | $N_2$ fixation contribution to PP (%) (Redfield 6.6 ratio) | SD | $N_2$ fixation contribution to PP (%) (mean POC/PN ratio of $6.3 \pm 1.1$) | SD |
|---|---|---|---|---|---|---|
| | **Bel-3** | 46.5 | **0** | - | **0** | - |
| | **Bel-5** | 45.3 | **0** | - | **0** | - |
| ENACWsp | **Bel-7** | 44.6 | **2** | 0.4 | **1** | 0.4 |
| | **Geo-21** | 46.5 | **1** | 0.02 | **1** | 0.0 |
| | **Bel-9** | 42.4 | **1** | 0.1 | **1** | 0.1 |
| | **Bel-11** | 40.7 | **28** | 1.9 | **25** | 1.8 |
| ENACWst | **Bel-13** | 38.8 | **25** | 1.3 | **23** | 1.2 |
| | **Geo-1** | 40.3 | **3** | 0.2 | **3** | 0.1 |
| | **Geo-2** | 40.3 | **3** | 0.1 | **3** | 0.1 |
| | **Geo-13** | 41.4 | **3** | 0.1 | **3** | 0.1 |

**Figure legends**

**Figure 1:** Location of sampling stations during the Belgica BG2014/14 (black labels) and GEOVIDE (white labels) cruises (May 2014) superimposed on a map of the seasonal average phosphate excess ($P^* = [PO_4^{3-}] - [NO_3^-] / 16$) at 20 m (April to June from 1955 to 2012; World Ocean Atlas 2013; Garcia et al., 2013). Areas of dominance of the Eastern North Atlantic Central Waters of subpolar (ENACWsp) and subtropical (ENACWst) origin are separated by an orange dashed line. Black dashed and solid contour lines illustrate 500 m and 1500 m isobaths, respectively. (Schlitzer, R., Ocean Data View).

**Figure 2:** θ/S diagrams obtained using CTD profiles down to 1500 m depth during **(a)** the Belgica BG2014/14 cruise (stations Bel-3, 5, 7, 11 and 13), **(b)** the GEOVIDE cruise (stations Geo-1, 2, 13 and 21) and **(c)** both expeditions combined. Diamonds indicate the characteristics of the major water masses encountered as reported in Fiúza (1984) and García-Ibáñez et al. (2015): Eastern North Atlantic Central Waters (ENACW) of subpolar (ENACWsp) and subtropical (ENACWst) origin, Mediterranean Water (MW) and Labrador Sea Water (LSW). (Schlitzer, R., Ocean Data View).

**Figure 3:** Spatial distribution of Chl $a$ **(a, d)**, $NH_4^+$ **(b, e)** and $NO_3^- + NO_2^-$ **(c, f)** concentrations along the Belgica BG2014/14 **(a to c)** and GEOVIDE **(d to f)** cruise tracks. Station numbers are indicated above the sections. The vertical black line represents the boundary between areas with dominance of Eastern North Atlantic Waters of subpolar (ENACWsp) and subtropical (ENACWst) origin. Mixed layer depth (MLD, black lines connecting diamonds) was estimated using a temperature threshold criterion of 0.2°C relative to the temperature at 10 m (de Boyer Montégut et al., 2004). (Schlitzer, R., Ocean Data View).

**Figure 4:** Spatial distribution ($\pm$ SD) of depth-integrated primary production (duplicates are in light and dark green with the corresponding bar values on top in mmol C m$^{-2}$ d$^{-1}$) determined during the **(a)** Belgica BG2014/14 and **(b)** GEOVIDE cruises. Error bars represent the propagated measurement uncertainty of all parameters used to compute volumetric uptake rates. Time series of area-averaged chlorophyll a concentration (mg m$^{-3}$) for the period between December 2013 and December 2014 for the 0.5° x 0.5° grid surrounding each sampled station during the **(c)** Belgica BG2014/14 and **(d)** GEOVIDE cruises, registered by Aqua MODIS satellite (Giovanni online satellite data system). Dashed box illustrated the sampling period for both cruises (May 2014).

**Figure 5:** Spatial distribution ($\pm$ SD) of depth-integrated $N_2$ fixation rates (duplicates are in light and dark blue with the corresponding bar values on top in µmol N m$^{-2}$ d$^{-1}$) determined during the **(a)** Belgica BG2014/14 and **(b)** GEOVIDE cruises. Error bars represent the propagated measurement uncertainty of all parameters used to compute volumetric uptake rates. Diversity of *nifH* sequences during **(c)** the Belgica BG2014/14 cruise, successfully recovered only at stations Bel-11 and Bel-13 (5 m), and during **(d)** the GEOVIDE cruise for stations Geo-2 (100 m), Geo-13 (35 m) and Geo-21. The total number of sequences recovered from each station sample is indicated on top of the bars, and the exact percentage represented by each group is shown inside the bars.

**Figure 6:** Euclidean distance biplot illustrating the axis loadings corresponding to the two components as obtained from the result of PCA based on Spearman rank correlation with depth-integrated rates of $N_2$ fixation and primary production (PP), phosphate excess (average $P^*$ at 20 m depth surrounding each sampled site from the April to June; World Ocean Atlas 2013 climatology from 1955 to 2012) (Garcia et al., 2013), average dust deposition (dry + wet)

derived during April 2014 satellite data (Giovanni online data system, NASA Goddard Earth Sciences Data and Information Services Center) and ambient variables (temperature, salinity, and nutrient data). Coloured dots represent the projection of each station corresponding to their biogeochemical characteristics. Axis 1 is found highly inversely correlated to PP, Chl $a$, $NH_4^+$ and $NO_3^- + NO_2^-$ concentrations, while highly positively related to temperature and $N_2$ fixation rates, with axis loading of –0.812, –0.768, –0.936, –0.783, 0.942 and 0.506, respectively (see table S5). Axis 2 is highly positively correlated to climatological P*, salinity and $N_2$ fixation rates (axis loading of 0.584, 0.943 and 0.602, respectively).

[Figure]

**Figure 1**

[Figure]

**Figure 2**

**Belgica BG2014/14**

Figure panels (a), (b), (c): Chl a (µg L⁻¹), NH₄⁺ (µmol L⁻¹), NO₃⁻ + NO₂⁻ (µmol L⁻¹)

**GEOVIDE**

Figure panels (d), (e), (f): Chl a (µg L⁻¹), NH₄⁺ (µmol L⁻¹), NO₃⁻ + NO₂⁻ (µmol L⁻¹)

**Figure 3**

[Figure]

**Figure 4**

[Figure]

[Figure]

**Figure 5**

[Figure]

**Figure 6**

Fonseca-Batista et al. manuscript revision

**Biogeosciences Editor's comments**

This study was previously submitted to Frontiers in Microbiology (FiTM) and was rejected after critical reviews by two referees. Both reviewers are positive about the publication of this study in FiTM, although the comments are pretty critical, particularly reviewer#2. So the reason of rejection appears to be that the authors are not able to adequately address the concerns raised by the reviewers. Both reviewers are nice, despite of being somehow critical.
I have carefully read the comments and replies to the comments, and reviewed the ms myself. The manuscript could be published only after major revision suggested by both reviewers.

**We would like to thank the Editor for taking the time to review this manuscript himself. We have carefully considered the comments and have revised the manuscript accordingly. We addressed all the comments in a point-by-point manner.**

The key concerns are from reviewer#2, and I fully side with him/her. "*Overall the text is too speculative- the discussion, although well written, is much too long with 10 pages, and especially the last 5 pages are full of speculation and partly wander too far from discussing actually measured data. It is impressive from how many different angles the authors try to shed light on their data, but when large parts of text end in hypothetical conclusions, assumptions, possibilities, it somehow makes the story implode back onto its core data, and it makes reading the long discussion a bit frustrating. I feel that the few interesting points could be summarized and put into necessary context in a much shorter way. I wonder whether the manuscript could be partly rewritten in order to focus more on the rates and less on the weak nifH data*".

**We understand the Editor's concerns related to the comments made by the Reviewer #2 during processing by Frontiers in Marine Science (FiMS). However, we would like to point out that significant modifications had been made in the manuscript prior to the submission to Biogeosciences, and further adjustments were now implemented to focus the text on our data and less on hypothetical environmental drivers:**

**(1) the discussion section is now 5 pages (compared to the initial 10 pages mentioned above). The most criticized section related to the environmental drivers of $N_2$ fixation now only has 2 pages (instead of 5).**

**(2) the main hypotheses put forward to explain the distribution and magnitude of $N_2$ fixation rates observed in this study were better summarized, made more concise and backed up with additional data from the GEOVIDE cruise that were not included in the last version submitted to FiMS. These supporting data include particulate organic carbon concentration in the surface waters and export efficiency (reflecting the potential for shallow recycling) (Lemaitre et al., under review in this same BG special issue for the GEOVIDE project), dissolved iron concentrations (Tonnard et al., under review in the GEOVIDE special issue), and *nifH* data on the diazotroph community in the Iberian Basin and West European Basin.**

**Please note that the *nifH* sequences recovered from the GEOVIDE sampling have recently been submitted to the GenBank database, and we will soon have the accession numbers for them. These will be added to the manuscript (methods section 2.3) and the phylogenetic tree in Supplementary Figure S1, before publication, if the manuscript is accepted.**
The major revisions should be made including:
(1) Stay focused on what you want to say. The authors need to focus on N2 fixation and productive water in this study, and get all relevant placed in the text first.
(2) The figures and tables need to be re-structured.

In summary, the major re-structuring is required as following.
   (1) Table 1 might be useful, but table 2 can be placed in the supplementary text.

**A major restructuring of the figures and tables was implemented as suggested by the Editor:**
- **both pair of Figures (Fig. 4-5 and Fig. 7-8) have now been merged.**
- **Fig. 6 was removed from the article, the phytoplankton pigments data is now only cited as part of another manuscript in preparation, as follows: Tonnard et al., in preparation.**
- **Table 2 was transferred to supplementary material.**

**Based on the BG Editor's latest comments, texts were adapted, particularly in the abstract, introduction, methods and results (to add DNA analysis and results from GEOVIDE cruise) and finally in the discussion to focus more on $N_2$ fixation rates, their significance at the regional and basin scales, and their contribution to primary production.**

The key points includes:
(1) N2 fixation rate should be the figure 2, and primary production figure 3. Other figures are then arranged in an order to explain the N2 fixation and primary production
(2) In addition, Fig.7 and Fig.8 can be merged. Figure 4 and Figure 5 might be merged as well.
(3) Figure 6 may not be appropriate in this study because it will be published somewhere else.

**Figures and Tables were re-arranged as suggested (see above). Concerning the order of figure presentation, we preferred discussing the sections on θ/S diagrams and nutrients and chlorophyll prior to the rates measurements. We believe that introducing the physico-chemical features of the studied region not only allows one to easily grasp the environmental context of the study, but also to highlight specific regions with characteristic traits of interest which will not need to be detailed when discussing $N_2$ fixation rates and Primary Production.**

Other key concerns for example are the following
(4) In the abstract, the table 1 was not mentioned which does not echo with the title "productive water"

**We agree, the abstract was adapted to focus more on $N_2$ fixation rates, their contribution to primary production, but also to reduce the attention brought to specific hypothetical environmental drivers.**

(5) Line 24 delete the phrase of (38.8–46.5° N; 8.0–19.7° W) in May 2014

**The latitudinal and longitudinal range of the studied region was deleted, we just kept "May 2014" to indicate the time of sampling which is relevant based on the importance of seasonality in this work, a condition now better highlighted in the abstract (line 23)**

(6) L26-28. This statement is too descriptive

**The sentence was modified, to bring the reader's attention to the facts that (1) such high $N_2$ fixation rates have not yet been observed earlier along the whole eastern Atlantic boundary, (2) elevated rates like these have only been reported at the western Atlantic boundary in very specific nutrient-rich environments such as: coastal, shelf and mesohaline waters. The sentence now reads as follows (lines 29-31):**

*"In the Atlantic Ocean, $N_2$ fixation rates exceeding 1000 µmol N $m^{-2}$ $d^{-1}$ have previously only been reported in the temperate and tropical western North Atlantic waters having coastal, shelf or mesohaline characteristics, as opposed to the mostly open ocean conditions studied here."*

(7) L28. Delete the phrase In agreement with previous studies,

**The sentence was deleted and the text was adapted as follows (lines 31-35 and 37-39):**

*"At the two sites where $N_2$ fixation activity was the highest; nifH sequences assigned to the prymnesiophyte-symbiont Candidatus Atelocyanobacterium thalassa (UCYN-A) dominated the nifH sequence pool recovered from DNA samples, while the remaining sequences, as for all the ones recovered from the other sites, belonged exclusively to non-cyanobacterial phylotypes."*
*…*
*"Earlier studies in the Iberian region were conducted largely outside the bloom period, unlike the present work which was carried out in spring, yet in all cases the assessment of nifH gene diversity, suggests a predominance of UCYN-A and non-cyanobacterial diazotrophs."*

(8) L35-38. Please provide solid evidence, instead of proposing something.

**The hypotheses are now introduced along with supporting information presented in the corresponding discussion section, regarding the abundance of particulate organic carbon in surface waters (based on Lemaitre et al., under review in the same GEOVIDE BG Special Issue), and on the in situ excess phosphorus data. The text was adapted as follows (lines 35-43):**

*"Previous studies in the Iberian Basin have systematically reported lower $N_2$ fixation rates (from < 0.1 to 140 µmol N $m^{-2}$ $d^{-1}$), as compared to those found in the present study, and this regardless of whether the bubble-addition method or the dissolution method were applied. Earlier studies in the Iberian region were conducted largely outside the bloom period, unlike the present work which was carried out in spring, yet in all cases the assessment of nifH gene diversity, suggests a predominance of UCYN-A and non-cyanobacterial diazotrophs. We support that the unexpectedly high $N_2$ fixation activities recorded at the time of our study were promoted by the availability of phytoplankton-derived organic matter produced during the spring bloom, as evidenced by the significant surface particulate organic carbon concentrations, and by the presence of excess phosphorus signature in surface waters, particularly at the sites with extreme activities."*

(9) L39-40. These data appear to be placed in the supplementary materials. Therefore, it is no appropriate to be discussed in the abstract.

**We agree, that sentence was deleted.**

**Biogeosciences Reviewer #1's comments**

Review of "Evidence of high N2 fixation rates in productive waters of the temperate Northeast Atlantic" by Fonseca-Batista et al.

The editor should understand that neither N fixation, N-fixing gene abundances, nor bioecology are within my realm of expertise. Despite that, I have sailed on many research cruises with N-fixation scientists, and even collected nifH samples for them on my own cruises. So I should be classified as a knowledgeable non-expert: I appreciate the research area but cannot analyze or critique details.

From that perspective, the paper is a fine contribution, calling attention to high N fixation rates and relatively high N-fixing gene copies in the temperate eastern Atlantic in the springtime following the spring bloom. This has not been observed before, partially due to methodological issues and partly due to the lack of observations in this season. Obviously it may change our view of how to assess global N fixation rates and how to model them.

**We would like to thank the Reviewer #1 for reviewing this manuscript even though the topic it covers are not exactly within his/her area of expertise. We have considered the comments, and we appreciate the Reviewer's recognition of our manuscript.**

I could only find a couple of small issues with the text (noted below), otherwise I believe it can be published with only minor revisions, hopefully with more guidance from a reviewer expert in this field.

(1) p. 1, line 31: For the sake of clarity, I recommend modifying the text to "At the sites where N2 fixation activity was the highest, we recovered sequences affiliated to UCYN-A1 (obligate symbiont of eukaryotic preymnesiophyte algae)."

**The sentence was modified and now reads as follows (lines 31-35):**

*"At the two sites where N$_2$ fixation activity was the highest; nifH sequences assigned to the prymnesiophyte-symbiont Candidatus Atelocyanobacterium thalassa (UCYN-A) dominated the nifH sequence pool recovered from DNA samples, while the remaining sequences, as for all the ones recovered from the other sites, belonged exclusively to non-cyanobacterial phylotypes."*

(2) p. 5, line 161: the Ambar and Fiúza, 1994 paper is not in the list of references.

**The missing reference was added to the reference list:**

*"Ambar, I., Fiúza, A.F.G. (1994). Some features of the Portugal Current System: a poleward slope undercurrent, an upwelling-related summer southward flow and an autumn-winter poleward coastal surface current. In: Proceedings of the Second International Conference on Air-Sea Interaction and on Meteorology and Oceanography of the Coastal Zone. Katsaros, K.B., Fiúza, A.F.G., Ambar, I., American Meteorological Society, pp. 286-287."*

*Supplementary Material*

**Evidence of high N₂ fixation rates in productive waters of the temperate Northeast Atlantic**

Debany Fonseca-Batista*, Xuefeng Li, Virginie Riou, Valérie Michotey, Florian Deman, François Fripiat, Sophie Guasco, Natacha Brion, Nolwenn Lemaitre, Manon Tonnard, Morgane Gallinari, Hélène Planquette, Frédéric Planchon, Géraldine Sarthou, Marc Elskens, Julie LaRoche, Lei Chou, Frank Dehairs

**\* Correspondence:**

Debany Fonseca-Batista:

dbatista8@hotmail.com

**Contents of this file**

Figures S1 to S3

Tables S1 to S4

[Figure]

**Supplementary Figure S1.** Phylogenetic tree of *nifH* predicted amino acid sequences generated using the Maximum Likelihood method of the Kimura 2-parameter model (Kimura, 1980) via the Molecular Evolutionary Genetics Analysis software (MEGA 7.0) (Kumar et al., 2016). Initial tree(s) for the heuristic search were obtained automatically by applying Neighbor-Join and BioNJ algorithms to a matrix of pairwise distances estimated using the Maximum Composite Likelihood (MCL) approach, and then selecting the topology with superior log likelihood value. A discrete Gamma distribution was used to model evolutionary rate differences among sites (5 categories (+G, parameter = 0.4038)). All sequences recovered from DNA samples, including those previously identified and the newly recovered ones (with ≥ 95% similarity at the nucleotide level with representative clones) are indicated with a blue rectangle. For the *nifH* sequences recovered from the GEOVIDE cruise, only those contributing to the cumulative 98% of recovered sequences were included in this tree. Bootstrap support values (≥ 50%) for 100 replications are shown at nodes. The scale bar indicates the number of sequence substitutions per site. The archaean *Methanobrevibacter smithii* was used as an outgroup. Accession numbers for published sequences used to construct the phylogenetic tree are given.

**Supplementary Table S1.** Summary of the dataset used to run the principal component analyses relating the volumetric rates of $N_2$ fixation and primary production to environmental variables

| Station | Date | Depth [m] | Lat. [°N] | Long. [°E] | $N_2$ fixation [nmol N $L^{-1}$ $d^{-1}$] | Primary production [μmol C $m^{-3}$ $d^{-1}$] | Temperature [°C] | Salinity [psu] | $NH_4^+$ [μM] | $NO_3^-+NO_2^-$ [μM] | P* [μM] | Chl $a$ [μg $L^{-1}$] |
|---------|------|-----------|-----------|------------|-----------|----------|----------|----------|----------|----------|----------|----------|
| Bel-3 | 24-May-2014 | 5 | 46.5 | -8.0 | < DL | 1180.8 | 13.86 | 35.58 | < DL | 0.27 | 0.04 | 1.42 |
| Bel-3 | 24-May-2014 | 10 | 46.5 | -8.0 | < DL | 776.1 | 13.87 | 35.58 | < DL | 0.36 | 0.04 | 1.25 |
| Bel-3 | 24-May-2014 | 25 | 46.5 | -8.0 | < DL | 995.9 | 13.78 | 35.58 | < DL | 1.14 | 0.06 | 1.10 |
| Bel-3 | 24-May-2014 | 50 | 46.5 | -8.0 | < DL | 109.7 | 12.44 | 35.62 | 0.35 | 4.62 | -0.03 | 0.16 |
| Bel-5 | 25-May-2014 | 5 | 45.3 | -8.8 | < DL | 429.1 | 13.91 | 35.59 | 0.19 | 0.66 | 0.02 | 0.16 |
| Bel-5 | 25-May-2014 | 30 | 45.3 | -8.8 | < DL | 360.6 | 13.91 | 35.59 | 0.59 | 0.91 | 0.00 | 0.12 |
| Bel-5 | 25-May-2014 | 70 | 45.3 | -8.8 | < DL | 433.9 | 13.24 | 35.61 | 0.59 | 4.71 | -0.04 | 0.21 |
| Bel-5 | 25-May-2014 | 130 | 45.3 | -8.8 | < DL | 9.7 | 11.92 | 35.62 | < DL | 7.74 | -0.14 | 0.00 |
| Bel-7 | 26-May-2014 | 5 | 44.6 | -9.3 | 1.1 | 849.1 | 13.94 | 35.51 | < DL | < DL | 0.05 | 1.19 |
| Bel-7 | 26-May-2014 | 16 | 44.6 | -9.3 | 1.0 | 707.3 | 13.94 | 35.52 | < DL | 0.24 | 0.05 | 1.14 |
| Bel-7 | 26-May-2014 | 30 | 44.6 | -9.3 | 2.0 | 1018.3 | 13.86 | 35.52 | 0.09 | 1.01 | 0.00 | 0.98 |
| Bel-7 | 26-May-2014 | 80 | 44.6 | -9.3 | 1.6 | 70.2 | 13.32 | 35.55 | 0.34 | 2.84 | 0.02 | 0.16 |
| Geo-21 | 31-May-2014 | 10 | 46.5 | -19.7 | 8.2 | 2824.8 | 14.57 | 35.68 | 0.18 | 1.52 | | 0.88 |
| Geo-21 | 31-May-2014 | 18 | 46.5 | -19.7 | < DL | 3443.7 | 13.70 | 35.69 | 0.39 | 2.21 | | 1.21 |
| Geo-21 | 31-May-2014 | 25 | 46.5 | -19.7 | 4.9 | 3500.1 | 13.47 | 35.68 | 0.50 | 2.82 | | 0.73 |
| Geo-21 | 31-May-2014 | 40 | 46.5 | -19.7 | 1.4 | 1155.1 | 12.84 | 35.65 | 0.68 | 4.13 | | 0.39 |
| Geo-21 | 31-May-2014 | 60 | 46.5 | -19.7 | 2.0 | 393.8 | 12.84 | 35.69 | 0.41 | 5.32 | | 0.19 |
| Geo-21 | 31-May-2014 | 91 | 46.5 | -19.7 | 2.3 | 73.8 | 12.51 | 35.70 | < DL | 7.19 | | |

**Supplementary Table S1 continued.**

| Station | Date | Depth [m] | Lat. [°N] | Long. [°E] | N$_2$ fixation [nmol N L$^{-1}$ d$^{-1}$] | Primary production [μmol C m$^{-3}$ d$^{-1}$] | Temperature [°C] | Salinity [psu] | NH$_4^+$ [μM] | NO$_3^-$+NO$_2^-$ [μM] | P* [μM] | Chl $a$ [μg L$^{-1}$] |
|---|---|---|---|---|---|---|---|---|---|---|---|---|
| **Bel-9** | 27-May-2014 | 5 | 42.4 | -9.7 | **3.9** | **335.9** | 16.04 | 35.55 | < DL | < DL | 0.05 | 0.16 |
| **Bel-9** | 27-May-2014 | 25 | 42.4 | -9.7 | **0.7** | **207.3** | 15.96 | 35.56 | < DL | 0.38 | 0.04 | 0.18 |
| **Bel-9** | 27-May-2014 | 45 | 42.4 | -9.7 | **0.9** | **571.2** | 14.33 | 35.83 | < DL | 2.07 | -0.01 | 0.55 |
| **Bel-9** | 27-May-2014 | 120 | 42.4 | -9.7 | **< DL** | **6.9** | 13.24 | 35.78 | < DL | 5.99 | -0.08 | 0.01 |
| **Bel-11** | 28-May-2014 | 5 | 40.7 | -11.1 | **65.4** | **565.3** | 16.95 | 35.56 | < DL | < DL | 0.05 | 0.15 |
| **Bel-11** | 28-May-2014 | 35 | 40.7 | -11.1 | **7.0** | **630.7** | 15.18 | 35.84 | < DL | < DL | 0.05 | 0.28 |
| **Bel-11** | 28-May-2014 | 45 | 40.7 | -11.1 | **< DL** | **292.3** | 15.26 | 35.83 | < DL | 0.61 | 0.02 | 0.12 |
| **Bel-11** | 28-May-2014 | 80 | 40.7 | -11.1 | **4.9** | **334.6** | 14.01 | 35.90 | < DL | 4.35 | -0.08 | 0.23 |
| **Bel-13** | 29-May-2014 | 5 | 38.8 | -11.4 | **45.0** | **599.4** | 17.23 | 35.68 | < DL | < DL | 0.05 | 0.22 |
| **Bel-13** | 29-May-2014 | 30 | 38.8 | -11.4 | **10.5** | **323.2** | 16.46 | 35.89 | < DL | < DL | 0.09 | 0.09 |
| **Bel-13** | 29-May-2014 | 45 | 38.8 | -11.4 | **12.6** | **692.0** | 15.38 | 35.97 | < DL | 0.52 | 0.12 | 0.20 |
| **Bel-13** | 29-May-2014 | 80 | 38.8 | -11.4 | **2.4** | **92.6** | 14.84 | 36.10 | < DL | 2.39 | 0.07 | 0.08 |
| **Geo-1** | 19-May-2014 | 6 | 40.3 | -10.0 | **4.8** | **621.8** | 16.70 | 35.11 | 0.33 | < DL | | 0.16 |
| **Geo-1** | 19-May-2014 | 11 | 40.3 | -10.0 | **7.1** | **696.9** | 16.53 | 35.18 | < DL | < DL | | 0.19 |
| **Geo-1** | 19-May-2014 | 16 | 40.3 | -10.0 | **4.8** | **667.8** | 16.09 | 35.26 | < DL | < DL | | |
| **Geo-1** | 19-May-2014 | 25 | 40.3 | -10.0 | **2.5** | **579.1** | 15.33 | 35.36 | < DL | 0.42 | | 0.33 |
| **Geo-1** | 19-May-2014 | 34 | 40.3 | -10.0 | **1.2** | **842.3** | 15.14 | 35.46 | < DL | 0.76 | | 0.35 |
| **Geo-1** | 19-May-2014 | 48 | 40.3 | -10.0 | **1.1** | **676.9** | 14.35 | 35.62 | < DL | 2.44 | | 0.46 |
| **Geo-2** | 20-May-2014 | 11 | 40.3 | -9.5 | **4.7** | **474.3** | 16.82 | 34.99 | < DL | < DL | | 0.21 |
| **Geo-2** | 20-May-2014 | 31 | 40.3 | -9.5 | **2.8** | **1170.5** | 14.67 | 35.54 | < DL | 0.93 | | 0.47 |
| **Geo-2** | 20-May-2014 | 39 | 40.3 | -9.5 | **2.7** | **1149.5** | 13.97 | 35.70 | 0.09 | 1.38 | | 0.98 |
| **Geo-2** | 20-May-2014 | 85 | 40.3 | -9.5 | **2.3** | **72.5** | 13.32 | 35.77 | < DL | 4.39 | | 0.02 |

**Supplementary Table S1 final.**

| Station | Date | Depth [m] | Lat. [°N] | Long. [°E] | N$_2$ fixation [nmol N L$^{-1}$ d$^{-1}$] | Primary production [μmol C m$^{-3}$ d$^{-1}$] | Temperature [°C] | Salinity [psu] | NH$_4^+$ [μM] | NO$_3^-$+NO$_2^-$ [μM] | P* [μM] | Chl $a$ [μg L$^{-1}$] |
|---|---|---|---|---|---|---|---|---|---|---|---|---|
| **Geo-13** | 24-May-2014 | 15 | 41.4 | -13.9 | **5.5** | **1670.7** | 15.47 | 35.84 | 0.21 | 0.34 | | 0.56 |
| **Geo-13** | 24-May-2014 | 30 | 41.4 | -13.9 | **1.0** | **403.0** | 14.73 | 35.81 | 0.34 | 0.68 | | 0.68 |
| **Geo-13** | 24-May-2014 | 43 | 41.4 | -13.9 | **2.4** | **910.9** | 13.66 | 35.77 | 0.73 | 2.21 | | |
| **Geo-13** | 24-May-2014 | 58 | 41.4 | -13.9 | **2.2** | **790.5** | 13.36 | 35.76 | 0.68 | 3.47 | | 0.45 |
| **Geo-13** | 24-May-2014 | 75 | 41.4 | -13.9 | **3.9** | **338.1** | 13.14 | 35.76 | 0.07 | 4.54 | | 0.17 |
| **Geo-13** | 24-May-2014 | 116 | 41.4 | -13.9 | **3.1** | **22.8** | 12.91 | 35.75 | < DL | 6.27 | | 0.01 |

**Supplementary Table S2.** Summary of the dataset used to run the principal component analyses relating the depth-integrated rates of $N_2$ fixation and primary production to environmental variables.

| Station | MLD | Lat. [°N] | Long. [°E] | $N_2$ fixation [µmol N m$^{-2}$ d$^{-1}$] | Primary Production [mmol C m$^{-2}$ d$^{-1}$] | Euphotic layer averaged Temperature [°C] | Euphotic layer averaged Salinity [psu] | Euphotic layer integrated Chl $a$ [mg m$^{-2}$] | Euphotic layer integrated NH$_4^+$ [µmol m$^{-2}$] | Euphotic layer integrated NO$_3^-$+NO$_2^-$ [µmol m$^{-2}$] | Climatology P* at 20 m depth [µmol m$^{-2}$] | Satellite-based dust deposition [Apr. 2014] [µg m$^{-2}$ d$^{-1}$] | Satellite-based dust deposition [May 2014] [µg m$^{-2}$ d$^{-1}$] |
|---|---|---|---|---|---|---|---|---|---|---|---|---|---|
| Bel-3 | 27 | 46.5 | -8.0 | 0 | 37.9 | 13.5 | 35.6 | 47.1 | 6.7 | 86.2 | 0.06 | 1263 | 2539 |
| Bel-5 | 41 | 45.3 | -8.8 | 0 | 42.6 | 13.2 | 35.6 | 17.1 | 54.0 | 509.0 | 0.06 | 1647 | 1914 |
| Bel-7 | 33 | 44.6 | -9.3 | 128 | 52.1 | 13.8 | 35.5 | 62.3 | 12.9 | 107.1 | 0.07 | 2147 | 1443 |
| Geo-21 | 17 | 46.5 | -19.7 | 279 | 135.3 | 13.3 | 35.7 | 38.1 | 34.2 | 452.5 | 0.04 | 2147 | 1443 |
| Bel-9 | 26 | 42.4 | -9.7 | 81 | 36.6 | 14.9 | 35.7 | 32.6 | 7.7 | 332.2 | 0.06 | 3650 | 1088 |
| Bel-11 | 33 | 40.7 | -11.1 | 1533 | 36.4 | 15.3 | 35.8 | 15.2 | 5.1 | 93.5 | 0.05 | 2799 | 618 |
| Bel-13 | 25 | 38.8 | -11.4 | 1355 | 35.9 | 16.0 | 35.9 | 12.1 | 5.1 | 58.3 | 0.06 | 2147 | 618 |
| Geo-1 | 12 | 40.3 | -10.0 | 141 | 33.1 | 15.7 | 35.3 | 14.2 | 5.3 | 68.4 | 0.03 | 3650 | 618 |
| Geo-2 | 14 | 40.3 | -9.5 | 262 | 59.1 | 14.7 | 35.5 | 37.8 | 6.0 | 219.4 | 0.01 | 4758 | 820 |
| Geo-13 | 29 | 41.4 | -13.9 | 384 | 78.9 | 13.9 | 35.8 | 42.4 | 34.0 | 274.2 | 0.07 | 1647 | 820 |

**Supplementary Table S3.** Spearman correlation matrix opposing depth-integrated rates of $N_2$ fixation and primary production (PP), from BG2014/14 and GEOVIDE cruises together, to euphotic layer averaged or integrated environmental variables. The correlation factor (r) and its significance given by the p-value (p) at $p < 0.001$, $p < 0.01$ and $p < 0.05$ presented with ***, ** and *, respectively, and the number of observations (n) are shown for each combination tested. Also, dFe correlations were made only for GEOVIDE sampling stations.

| | | Average salinity | Integrated $[NH_4^+]$ [µmol m$^{-2}$] | Integrated $[NO_3^- + NO_2^-]$ [µmol m$^{-2}$] | Average in situ P* [µM] | Surface climatology P* [µM] | Integrated dFe [nmol m$^{-2}$] | Satellite-based dust deposition April 2014 [µg m$^{-2}$ s$^{-1}$] | Satellite-based dust deposition May 2014 [µg m$^{-2}$ s$^{-1}$] | PP [mmol C m$^{-2}$ d$^{-1}$] | $N_2$ fixation [µmol N m$^{-2}$ d$^{-1}$] |
|---|---|---|---|---|---|---|---|---|---|---|---|
| Average temperature [°C] | r | 0.188 | -0.869*** | -0.721* | 0.0788 | -0.0732 | 0.8 | 0.587 | -0.901*** | -0.721* | 0.553 |
| | p | 0.583 | 0.0000002 | 0.0157 | 0.811 | 0.811 | 0.333 | 0.0665 | 0.0000002 | 0.0157 | 0.0892 |
| | n | 10 | 10 | 10 | 10 | 10 | 4 | 10 | 10 | 10 | 10 |
| Average salinity | | | -0.109 | 0.152 | 0.00606 | 0.366 | -0.6 | -0.29 | -0.265 | -0.0182 | 0.62* |
| | | | 0.733 | 0.656 | 0.973 | 0.275 | 0.417 | 0.404 | 0.446 | 0.946 | 0.048 |
| | | | 10 | 10 | 10 | 10 | 4 | 10 | 10 | 10 | 10 |
| Integrated $[NH_4^+]$ [µmol m$^{-2}$] | | | | 0.857*** | -0.0304 | 0.26 | -0.8 | -0.471 | 0.737* | 0.729* | -0.482 |
| | | | | 0.0000002 | 0.919 | 0.446 | 0.333 | 0.16 | 0.0131 | 0.0131 | 0.148 |
| | | | | 10 | 10 | 10 | 4 | 10 | 10 | 10 | 10 |
| Integrated $[NO_3^- + NO_2^-]$ [µmol m$^{-2}$] | | | | | -0.0182 | 0.116 | -0.8 | -0.228 | 0.512 | 0.758** | -0.195 |
| | | | | | 0.946 | 0.733 | 0.333 | 0.512 | 0.116 | 0.0087 | 0.559 |
| | | | | | 10 | 10 | 4 | 10 | 10 | 10 | 10 |
| Average in situ P* [µM] | | | | | | -0.274 | -0.4 | 0.161 | -0.253 | 0.479 | 0.511 |
| | | | | | | 0.425 | 0.75 | 0.631 | 0.468 | 0.148 | 0.116 |
| | | | | | | 10 | 4 | 10 | 10 | 10 | 10 |
| Surface climatology P* [µM] | | | | | | | -0.8 | -0.491 | 0.205 | 0.0732 | -0.0856 |
| | | | | | | | 0.333 | 0.137 | 0.559 | 0.811 | 0.785 |
| | | | | | | | 4 | 10 | 10 | 10 | 10 |
| Integrated dFe [nmol m$^{-2}$] | | | | | | | | 0.8 | -0.632 | -0.8 | -0.6 |
| | | | | | | | | 0.333 | 0.333 | 0.333 | 0.417 |
| | | | | | | | | 4 | 4 | 4 | 4 |
| Satellite-based dust deposition April 2014 [µg m$^{-2}$ s$^{-1}$] | | | | | | | | | -0.588 | -0.265 | 0.257 |
| | | | | | | | | | 0.0665 | 0.446 | 0.446 |
| | | | | | | | | | 10 | 10 | 10 |
| Satellite-based dust deposition May 2014 [µg m$^{-2}$ s$^{-1}$] | | | | | | | | | | 0.512 | -0.774** |
| | | | | | | | | | | 0.116 | 0.00686 |
| | | | | | | | | | | 10 | 10 |
| PP [mmol C m$^{-2}$ d$^{-1}$] | | | | | | | | | | | -0.0365 |
| | | | | | | | | | | | 0.892 |
| | | | | | | | | | | | 10 |

**Supplementary Table S4.** Spearman correlation matrix opposing volumetric rates of $N_2$ fixation and primary production (PP) to depth-specific environmental variables for the combined Belgica 2014/14 and GEOVIDE cruises. The correlation factor (r) and its significance given by the p-value *(p)* at $p < 0.001$, $p < 0.01$ and $p < 0.05$ presented with ***, ** and *, respectively, and the number of observations (n) are shown for each combination tested. Note that when nutrient concentrations were < DL we used the DL value to run the correlation test. Also, P* correlations were only made for the Belgica 2014/14 studied sites and dFe correlations only for GEOVIDE sampling stations.

| | | Salinity | $[NH_4^+]$ $[\mu M]$ | $[NO_3^- + NO_2^-]$ $[\mu M]$ | In situ P* $[\mu M]$ | DFe [nM] (Tonnard et al., 2018) | Chl *a* $[\mu g\ L^{-1}]$ | PP $[\mu mol\ C\ m^{-3}\ d^{-1}]$ | $N_2$ fixation $[nmol\ N\ L^{-1}\ d^{-1}]$ |
|---|---|---|---|---|---|---|---|---|---|
| Temperature | **r** | **-0.191** | **-0.418**\*\* | **-0.854**\*\*\* | **0.628**\*\* | **0.291** | **0.00461** | **0.158** | **0.54**\*\*\* |
| [°C] | *p* | *0.202* | *0.00398* | *0.0000002* | *0.00104* | *0.267* | *0.976* | *0.292* | *0.00012* |
| | n | 46 | 46 | 46 | 24 | 16 | 43 | 46 | 46 |
| Salinity | | | **0.0922** | **0.39**\*\* | **0.0288** | **0.0883** | **-0.195** | **-0.214** | **0.106** |
| | | | *0.54* | *0.00765* | *0.892* | *0.738* | *0.208* | *0.152* | *0.482* |
| | | | 46 | 46 | 24 | 16 | 43 | 46 | 46 |
| $[NH_4^+]$ | | | | **0.344**\* | **-0.441**\* | **-0.462** | **0.128** | **0.255** | **-0.154** |
| $[\mu M]$ | | | | *0.0194* | *0.031* | *0.0695* | *0.411* | *0.0869* | *0.307* |
| | | | | 46 | 24 | 16 | 43 | 46 | 46 |
| $[NO_3^- + NO_2^-]$ | | | | | **-0.716**\*\*\* | **-0.253** | **-0.216** | **-0.314**\* | **-0.362**\* |
| $[\mu M]$ | | | | | *0.0000325* | *0.336* | *0.162* | *0.0338* | *0.0136* |
| | | | | | 24 | 16 | 43 | 46 | 46 |
| In situ P* | | | | | | **undefined** | **0.149** | **0.369** | **0.557**\*\* |
| $[\mu M]$ | | | | | | *undefined* | *0.481* | *0.0746* | *0.00481* |
| | | | | | | 0 | 24 | 24 | 24 |
| DFe [nM] (Tonnard et al., 2018) | | | | | | | **-0.57** | **-0.416** | **0.175** |
| | | | | | | | *0.0322* | *0.104* | *0.505* |
| | | | | | | | 14 | 16 | 16 |
| Chl *a* $[\mu g\ L^{-1}]$ | | | | | | | | **0.85**\*\*\* | **-0.0874** |
| | | | | | | | | *0.0000002* | *0.575* |
| | | | | | | | | 43 | 43 |
| PP $[\mu mol\ C\ m^{-3}\ d^{-1}]$ | | | | | | | | | **0.143** |
| | | | | | | | | | *0.342* |
| | | | | | | | | | 46 |

[Figure]

**Supplementary Figure S2.** High resolution trihourly averaged sea surface temperature (SST) on May 29th 2014 (sampling date of station Bel-13, 10 days following the sampling of station Geo-1), derived from the European Sea (sea surface subskin temperature) of Copernicus Marine Environment Monitoring Service (CMEMS, European Commission). White markers indicate the location of the stations sampled during our study (image provided by Google Earth Pro).

[Figure]

[Figure]

**Supplementary Figure S3.** April (A) and May (B) 2014 monthly averaged dry + wet dust deposition (kg m$^{-2}$ s$^{-1}$) derived from Giovanni online satellite data system (NASA Goddard Earth Sciences Data and Information Services Center). White markers indicate the location of the stations sampled during our study (image provided by Google Earth Pro).

**Supplementary Table S5.** Principal component matrix illustrating the components (or axis) loadings, in other words the correlation of each variable to a determined axis as obtained with the XLSTAT software. The percentage of variability of the system explained by each of the two axes is indicated, for a total explained variance of 68%.

| | Axis 1 | Axis 2 |
|---|---|---|
| % Variability explained: | 48% | 20% |
| **Variables** | | |
| Euphotic layer integrated primary production | **-0.812** | 0.088 |
| Euphotic layer averaged temperature | **0.942** | 0.130 |
| Euphotic layer integrated Chl *a* | **-0.768** | -0.085 |
| Euphotic layer integrated [$NH_4^+$] | **-0.936** | -0.007 |
| Euphotic layer integrated [$NO_3^- + NO_2^-$] | **-0.783** | 0.154 |
| Climatology surface P* (20 m) | -0.305 | **0.584** |
| Euphotic layer averaged salinity | 0.125 | **0.943** |
| Satellite dry + wet dust deposition (April 2014) | 0.583 | -0.423 |
| **Euphotic layer integrated N$_2$ fixation** | **0.506** | **0.602** |

---

## Author Comment (AC3) · 24 Sep 2018

Dear Editor,

I failed to mention in the plain text where I respond to your comments that I also attached a supplement pdf file that includes the revised version of the manuscript, the new figure arrangements, a clearer version of the responses to your comments and those from the Reviewer #1, and the new Supplementary Material.

Please find this supplement pdf file attached to this additional response item.

Best regards,

[Figure]

Debany Fonseca P. Batista

Please also note the supplement to this comment:
https://www.biogeosciences-discuss.net/bg-2018-220/bg-2018-220-AC3-
supplement.pdf

―――――――――――――――――――――

---

## Author Response (AR1)

Dear Mr. Debany Fonseca-Batista,

I have again reviewed your ms on my own and unfortunately it cannot be published at its present form.

As you may still remember the key problem of your manuscript is the lack of logics, although the data have potential of publication.

PLEASE STAY FOCUSE on what you wanna convey as stated in the title: "Evidence of high N2 fixation rates in productive waters of the temperate Northeast Atlantic". Based on this title, the authors would expect to see the direct evidence for (1) N2 fixation rate; (2) productive water, i.e., 13C inorganic uptake; AND the reason behind it including (3) water nutrient and pigement; (4) taxonomic identity of N fixer; (4) environmental force determining the N2 fixation and inorganic 13C uptake.

Therefore, in the abstract, it is important to show your key findings in a logic way. i.e., you have observed N2 fixation at 8 stations out of 10 sampled; you also need to show the flux and importance of inorganic 13C uptake. Then you might explain what environmental variables play a key role in regulating N2 fixation and 13CO3 uptake, in addition to the taxonomic identity of N2 fixation.
Meanwhile, you could explain why the extraordinarily high rate of N2 fixation occurred.

**Dear Editor, we thank you for the time and work you have been putting in order to improve the reading of our manuscript. In order to take into account the above general comments you have brought to our attention, we first decided to adapt the title of the manuscript, which now reads as "Evidence of high N$_2$ fixation rates in the temperate Northeast Atlantic". Secondly, we incremented the abstract with more details about our findings in terms of primary production at the time of our study, while leaving out the information about previous studies in the same region (for N$_2$ fixation rate measurements and diazotrophs community assessment).**

Your current abstract cannot be published, and please first revise the Result section (figures and tables), and then materials and methods, then abstract.

**With regards to the Material and Methods and Results sections, we have modified the text in order to comply with the Editor's general views. Details about these adaptations are given below.**

Once again, I would like to emphasize that your ms has potential of publication, but it should be organized in a straightforward manner regarding the key findings you want to show.
If you do not want spend more time on it, it is also acceptable that you can notify the editorial office to withdraw this BG submission, and then seek for publication of this study somewhere else.

Best wishes
Zhongjun Jia
* * *
Comments to bg-2018-220
**Abstract**
(1) L26. Please delete the following phrase. (65 and 45 nmol N L-1 d-1 at surface level, respectively).

**The sentence was deleted**

(2) L27. Please delete the following sentence. "Although diazotrophic activity was not detected at two northern stations in the central Bay of Biscay". It is not necessary to emphasize these negative two sites. The authors can simply focus on the 8 sites where N2 fixation occurred during the sampling period.

**The sentence was deleted**

(3) L28. Might contribute to 1-3% f euphotic layer daily PP

**The sentence was adapted as follows (lines 30 to 32):**

*"When converted to carbon uptake using Redfield stoichiometry, $N_2$ fixation rates could have supported 1 to 3% of euphotic layer daily PP at most sites, except at the two most active sites where this contribution to daily PP reached as high as 25%".*

(4) L29-32. Pls delete the following sentence. In the Atlantic Ocean, N2 fixation rates exceeding 1000 µmol N m-2 d-1 have previously only been reported in the temperate and tropical western North Atlantic waters having coastal, shelf or mesohaline characteristics, as opposed to the mostly open ocean conditions studied here.

**The sentence was deleted**

(5) L35-37. Please delete the description of the early study, while emphasize your own results.

**The sentence was deleted**

(6) L37-40. Please rephrase the following statement, and delete the description of early study. Earlier studies in the Iberian region were conducted largely outside the bloom period, unlike the present work which was carried out in spring, yet in all cases the assessment of *nifH* gene diversity, suggests a predominance of UCYN-A and non-cyanobacterial diazotrophs.

**The sentence was deleted**

Materials and Methods. It should be re-structured as following.
(1) 2.1. Site description and water sample collection
(2) 2.2. N2 fixation Measurement (3) 2.3. inorganic uptake determination
(4) 2.3. Physiochemical and biological properties of oceanic water
(5) 2.4. DNA extraction and illumine sequencing of nifH genes
(6) 2.5. Statistical analysis

**The Material and Methods section was adapted in order to fit as closely as possible to the recommendation of the Editor. However, in order to provide the reader with a more linear presentation, as commonly done in oceanographic publications, the methods for nutrient measurements were described just after the site description and sample collection sub-section. Stable isotope incubation experiments ($^{15}$N and $^{13}$C) were described within a single sub-section, this particularly because each incubation bottle was spiked with both tracers. This way of presentation is the most commonly used in the literature. The Material and Methods section is now organized as follows:**

> **2.1 Site description and sample collection**
> **2.2 Nutrient measurements**
> **2.3 $^{15}$N$_2$ fixation and $^{13}$C-HCO$_3^-$ uptake rates**
> **2.4 DNA sampling and *nifH* diversity analysis**
> **2.5 Statistical analysis**

In addition
(7) The present writing could be improved for reading with great ease. The authors need to specifically describe how the water were sampled for determination of various properties. For example, L107, where the 12 or 14 Niskin bottles were placed (at depth) for seawater sampling?

**The text within the sub-section "Site description and sample collection" was adapted as follows (lines 111 to 118):**

*"Temperature, salinity and photosynthetically active radiation (PAR) profiles down to 1500 m depth were obtained using a conductivity-temperature-depth sensor (SBE 09 and SBE 911+, during the BG2014/14 and GEOVIDE cruises, respectively) fitted to rosette frames. For all biogeochemical measurements seawater samples were collected from Niskin bottles attached to the rosette and triggered at specific depths in the upper 200 m. In particular, for stable isotope incubation experiments seawater was collected in 4.5 L acid-cleaned polycarbonate (PC) bottles from four depths corresponding to 54%, 13%, 3% and 0.2% of surface PAR at stations Bel-3, 5, 7, 9, 11, and Geo-2. At stations Geo-1, 13 and 21, two additional depths corresponding to 25% and 1% of surface PAR were also sampled for the same purpose".*

(8) The methodological description for environmental condition assessment (such as nitrate measurment) should be placed after the description of sample collection

**The methodology for nutrient measurements is described in the sub-section following the site description and sample collection.**

(9) L121-124. This part could be placed in the section 2.1 as sample collection for N2 fixation and primary production

**The sentence was moved to the sample collection sub-section, as cited above (lines 115 to 118).**

(10) L130. 13C-HCO2 spiking could be placed in a separate paragraph.

**Nitrogen and carbon stable isotope incubations were carried out simultaneously in the same incubation bottles, this is now clarified in the text as follows (lines 133 to 135):**
*"$N_2$ fixation and primary production (PP) were determined simultaneously from the same incubation sample in duplicate using the $^{15}$N-$N_2$ dissolution method (Großkopf et al., 2012) and $^{13}$C-NaHCO$_3$ tracer addition (Hama et al., 1983) techniques, respectively".*

**This is why we find it appropriate to keep the description of the whole incubation experiment in a single sub-section.**

Results. It should be re-structured as following.
(7) 3.1. Inorganic 13C assimilation and N2 fixation. NEW figure 2, it can be made by combining Fig. 4ab and Fig. 5ab in the original ms as NEW Figure 2abcd. In this section, please describe the 13C changes of organic matter, instead of using the term primary production. Or you may start the paragraph by saying that the primary production was assessed by the increase of 13C in organic carbon
(8) 3.2. Water nutrients and pigment distribution. NEW Figure 3. It can be made by combining Figure 3 and Figure 4c and Figure 4d in the original ms.
(9) 3.3. Taxonomic identities of N2 fixers. NEW Figure 4. It is the supplementary Figure S1.
(10) 3.4. Environmental determinants of N2 fixation in productive water. NEW Figure 5. i.e. the Figure 6 in the original manuscript.

**We thank the Editor for the suggestions to reorganize the Results section. We have undertaken most of the changes pointed out by the Editor.**

➢ **Changes in the figure arrangements:**

**The new Fig. 3 combines the primary production rates and the $N_2$ fixation rates.**
**The phylogenetic tree that was previously in the supplemental information is now Fig. 5.**
**The water mass diagrams (prior Fig. 2) have been moved to the supplemental information.**

➢ **Changes to the text:**

• **To comply with the approach commonly used in the oceanographic literature, the Result section first describes the environmental context of our study by presenting the nutrient levels, and related biomass (chl *a*). As such, we kept nutrient and chlorophyll sections as in Fig. 2.**
• **Given the new structure of the result presentation, we have not merged the nutrient and chlorophyll sections with the satellite-based time series data, since the latter are discussed in the primary production section. The satellite-based chl *a* data is used in the manuscript to support our**

**primary production rate measurements, and to relate them to the stage of the spring bloom found at the different sites studied.**

- **The term "primary production" is now clearly define at the beginning of the Results' sub-section 3.2 (lines 227 to 230) which is now entitled "Primary production and satellite-based Chl *a* observations. In addition we have now added a paragraph in the methods describing the calculations, which are also clearly outlined in the reference cited in this section: Hama et al. (1983) (lines 150 to 154).**

**In the methods (lines 150 to 154):**
*"N$_2$ fixation and carbon uptake volumetric rates were computed as shown in Equation 1:*

$$N_2 \text{ or } HCO_3^- \text{ uptake rate } (\text{nmol } L^{-1}d^{-1}\text{or } \mu\text{mol } m^{-3}d^{-1}) = \frac{A_{PN \text{ or } POC}^{final} - A_{PN \text{ or } POC}^{t=0}}{A_{N_2 \text{ or } DIC} - A_{PN \text{ or } POC}^{t=0}} \times \frac{[PN \text{ or } POC]}{\Delta t} \qquad \textbf{(1)}$$

*where A represents the $^{15}$N or $^{13}$C atom% excess of PN or POC at the beginning (t =0) and end (final) of the incubation, or of the dissolved inorganic pool (N$_2$ or dissolved inorganic carbon, DIC); and Δt the incubation period".*

*Hama, T., Miyazaki, T., Ogawa, Y., Iwakuma, T., Takahashi, M., Otsuki, A., & Ichimura, S. (1983). Measurement of photosynthetic production of a marine phytoplankton population using a stable 13C isotope. Marine Biology, 73, 31–36.*

**In the Results (lines 227 to 230):**
*"Primary production (PP), estimated through the incorporation of enriched bicarbonate ($^{13}$C-NaHCO$_3$) into the particulate organic carbon (POC) pool, illustrated volumetric rates ranging from 7 to 3500 μmol C m$^{-3}$ d$^{-1}$ (see Supporting Information Table S1) and euphotic layer integrated rates ranging from 32 to 137 mmol C m$^{-2}$ d$^{-1}$ (Fig. 3a, b, and Supporting Information Table S2)".*

- **N$_2$ fixation rates were presented in the last sub-section of the Results, along with the taxonomic affiliation of the diazotrophs to support our observations, including the phylogenetic tree (new Fig. 6).**

- **Finally, a sub-section presenting the major findings related to statistical analyses (Spearman rank correlation) has been added at the end of the Results section, as suggested by the Editor (lines 280 to 287).**

The figure 2 in the current ms about water mass could be placed in the supplementary section.

**The potential temperature versus salinity diagrams (previously Fig. 2) were moved to the Supplementary Material as Fig. S1, as proposed.**

[revised manuscript text omitted]

**Figure 4:** ~~Spatial distribution (± SD) of depth integrated primary production (duplicates are in light and dark green
with the corresponding bar values on top in mmol C m$^{-2}$ d$^{-1}$) determined during theBelgica BG2014/14 and~~ **(b)**
~~GEOVIDE cruises. Error bars represent the propagated measurement uncertainty of all parameters used to compute
volumetric uptake rates.~~ Time series of area-averaged chlorophyll a concentration (mg m$^{-3}$) registered by Aqua
MODIS satellite (Giovanni online satellite data system) between December 2013 and December 2014 for the 0.5° x
0.5° grid surrounding the different stations during the (**a**) Belgica BG2014/14 and (**d**) GEOVIDE cruises. The
dashed box highlights the sampling period for both cruises (May 2014).

**Figure 5:** ~~Spatial distribution (± SD) of depth integrated N$_2$ fixation rates (duplicates are in light and dark blue with
the corresponding bar values on top in µmol N m$^{-2}$ d$^{-1}$) determined during theBelgica BG2014/14 and~~ **(b)**
~~GEOVIDE cruises. Error bars represent the propagated measurement uncertainty of all parameters used to compute
volumetric uptake rates.e~~)   the Belgica BG2014/14 cruise (successfully
recovered only at stations Bel-11 and Bel-13, 5 m) and (**b**) the GEOVIDE cruise (stations Geo-2, 100 m; Geo-13,
m and Geo-21, 15 and 70 m. The total numbers of recovered sequences are indicated on top of the bars, and the
exact percentage represented by each group is shown inside the bars.

**Figure 6:** Phylogenetic tree of *nifH* predicted amino acid sequences generated using the Maximum Likelihood
method of the Kimura 2-parameter model (Kimura, 1980) via the Molecular Evolutionary Genetics Analysis software
(MEGA 7.0) (Kumar et al., 2016). Initial tree(s) for the heuristic search were obtained automatically by applying
Neighbor-Join and BioNJ algorithms to a matrix of pairwise distances estimated using the Maximum Composite

Likelihood (MCL) approach, and then selecting the topology with superior log likelihood value. A discrete Gamma distribution was used to model evolutionary rate differences among sites (5 categories (+G, parameter = 0.4038)). All sequences recovered from DNA samples, including those previously identified and the newly recovered ones (with ≥ 95% similarity at the nucleotide level with representative clones) are highlighted in blue. For the *nifH* sequences recovered from the GEOVIDE cruise, only those contributing to the cumulative 98% of recovered sequences were included in this tree. Bootstrap support values (≥ 50%) for 100 replications are shown at nodes. The scale bar indicates the number of sequence substitutions per site. The archaean *Methanobrevibacter smithii* was used as an outgroup. Accession numbers for published sequences used to construct the phylogenetic tree are given.

**Figure 7:** Euclidean distance biplot illustrating the axis loadings for the two main PCA components based on the Spearman rank correlation matrix shown in Table S3. Variables taken into account include depth-integrated rates of $N_2$ fixation and primary production (PP), average phosphate excess at 20 m depth surrounding each sampled site recovered from World Ocean Atlas 2013 climatology data between April and June from 1955 to 2012 (Garcia et al., 2013); satellite average dust deposition (dry + wet) derived during April 2014 (Giovanni online data system, NASA Goddard Earth Sciences Data and Information Services Center) and ambient variables (temperature, salinity, and nutrient data). Coloured dots in the biplot represent the projection of the different stations. Axis 1 has high negative loadings for PP, Chl *a*, $NH_4^+$ and $NO_3^- + NO_2^-$, and high positive loadings for temperature and $N_2$ fixation rates, with values of –0.812, –0.768, –0.936, –0.783, 0.942 and 0.506, respectively (see table S5). Axis 2 has high positive loadings of 0.584, 0.943 and 0.602 for climatological P*, salinity and $N_2$ fixation rates, respectively. PCA analysis was run in XLSTAT 2017 (Addinsoft, Paris, France, 2017).

---

## Editor Decision (ED1)

Dear Mr. Debany Fonseca-Batista,

I have again reviewed your ms on my own and unfortunately it cannot be published at its present form.

As you may still remember the key problem of your manuscript is the lack of logics, although the data have potential of publication.

PLEASE STAY FOCUSE on what you wanna convey as stated in the title: "Evidence of high N2 fixation rates in productive waters of the temperate Northeast Atlantic". Based on this title, the authors would expect to see the direct evidence for (1) N2 fixation rate; (2) productive water, i.e., 13C inorganic uptake; AND the reason behind it including (3) water nutrient and pigement; (4) taxonomic identity of N fixer; (4) environmental force determining the N2 fixation and inorganic 13C uptake.

Therefore, in the abstract, it is important to show your key findings in a logic way. i.e., you have observed N2 fixation at 8 stations out of 10 sampled; you also need to show the flux and importance of inorganic 13C uptake. Then you might explain what environmental variables play a key role in regulating N2 fixation and 13CO3 uptake, in addition to the taxonomic identity of N2 fixation. Meanwhile, you could explain why the extraordinarily high rate of N2 fixation occurred.

Your current abstract cannot be published, and please first revise the Result section (figures and tables), and then materials and methods, then abstract.

Once again, I would like to emphasize    that your ms has potential of publication, but it should be organized in a straightforward manner regarding the key findings you want to show.

If you do not want spend more time on it, it is also acceptable that you can notify the editorial office to withdraw this BG submission, and then seek for publication of this study somewhere else.

Best wishes

Zhongjun Jia
* * *
Comments to bg-2018-220

**Abstract**
(1) L26. Please delete the following phrase. (65 and 45 nmol N $L_{-1}$ $d_{-1}$ at surface level, respectively).
(2) L27. Please delete the following sentence. "Although diazotrophic activity was not detected at two northern stations in the central Bay of Biscay". It is not necessary to emphasize these negative two sites. The authors can simply focus on the 8 sites where N2 fixation occurred during

the sampling period.

(3) L28. Might contribute to 1-3% f euphotic layer daily PP

(4) L29-32. Pls delete the following sentence. In the Atlantic Ocean, $N_2$ fixation rates exceeding 1000 µmol N $m_{-2}$ $d_{-1}$ have previously only been reported in the temperate and tropical western North Atlantic waters having coastal, shelf or mesohaline characteristics, as opposed to the mostly open ocean conditions studied here.

(5) L35-37. Please delete the description of the early study, while emphasize your own results.

(6) L37-40. Please rephrase the following statement, and delete the description of early study. Earlier studies in the Iberian region were conducted largely outside the bloom period, unlike the present work which was carried out in spring, yet in all cases the assessment of *nifH* gene diversity, suggests a predominance of UCYN-A and non-cyanobacterial diazotrophs.

Materials and Methods. It should be re-structured as following.

(1) 2.1. Site description and water sample collection

(2) 2.2. N2 fixation Measurement

(3) 2.3. inorganic uptake determination

(4) 2.3. Physiochemical and biological properties of oceanic water

(5) 2.4. DNA extraction and illumine sequencing of nifH genes

(6) 2.5. Statistical analysis

In addition

(7) The present writing could be improved for reading with great ease. The authors need to specifically describe how the water were sampled for determination of various properties. For example, L107, where the 12 or 14 Niskin bottles were placed (at depth) for seawater sampling?

(8) The methodological description for environmental condition assessment (such as nitrate measurment) should be placed after the description of sample collection

(9) L121-124. This part could be placed in the section 2.1 as sample collection for N2 fixation and primary production

(10) L130. 13C-HCO2 spiking could be placed in a separate paragraph.

Results. It should be re-structured as following.

(7) 3.1. Inorganic 13C assimilation and N2 fixation. NEW figure 2, it can be made by combining Fig. 4ab and Fig. 5ab in the original ms as NEW Figure 2abcd. In this section, please describe the 13C changes of organic matter, instead of using the term primary production. Or you may start the paragraph by saying that the primary production was assessed by the increase of 13C in organic carbon

(8) 3.2. Water nutrients and pigment distribution. NEW Figure 3. It can be made by combining Figure 3 and Figure 4c and Figure 4d in the original ms.

(9) 3.3. Taxonomic identities of N2 fixers. NEW Figure 4. It is the supplementary Figure S1.

(10) 3.4. Environmental determinants of N2 fixation in productive water. NEW Figure 5. i.e. the Figure 6 in the original manuscript.

The figure 2 in the current ms about water mass could be placed in the supplementary section.

---

## Author Response (AR2)

**Associate Editor Decision: Publish subject to minor revisions (review by editor)** (21 Jan 2019) by Zhongjun Jia
Comments to the Author:

Dear Mr. Debany Fonseca-Batista

Thank you for your submission to BG. I have read your manuscript and found the major concerns have been addressed.

**To ensure the reproducibility of N2 fixation and carbon uptake measurements, the methods need to be described in greater detail. However, I could not see the methodological details in section 3.3 as stated in L161 as stated. Please describe it as much as possible in the materials and method sections rather than in the result section.**

We thank the Editor for pointing out this aspect as well as the following minor concerns. We have modified the methodological section to describe in more details the incubation experiments (lines 136 to 168). We have also listed in the method section the incubation that did not reveal any detectable $N_2$ fixation activity, as a result of an the insufficient or absent $^{15}N$-enrichment of the particles after incubation (lines 179 to 181).

Concerning the more specific comments please find the detailed modifications made below.

**The language must be polished by a native English speaker before your next submission.**

The manuscript has been read by a native English speaker, once all other requested revisions were made.

**Some other minor concerns are the following.**
**(1) L23. Delete "we report", and rephrased as "Substantial N2 fixation activity was observed at ...**

The sentence was modified accordingly.

**(2) L28-29. Along with the area-averaged Chl a concentrations, these results revealed that post-bloom prevailed at most sites....**

The sentenced was adapted as follows (lines 27 to 29):

*"In agreement with the area-averaged Chl a satellite data contemporaneous with our study period, our results revealed that post-bloom conditions prevailed at most sites, while at the northwesternmost station the bloom was still ongoing."*

**(3) L31. Delete "we find that"**

The sentenced was adapted as follows (lines 29 to 31):

*"When converted to carbon uptake using Redfield stoichiometry, $N_2$ fixation could support 1 to 3% of daily PP in the euphotic layer at most sites, except at the two most active sites where this contribution to daily PP could reach up to 25%."*

**(4) L33. nifH sequences were assigned to**
**(5) L34 that dominated nifH sequence.**
**(6) L34 delete "recovered from DNA samples"**

The sentenced was adapted as follows (lines 31 to 34):

*"At the two sites where N₂ fixation activity was the highest, the prymnesiophyte-symbiont Candidatus Atelocyanobacterium thalassa (UCYN-A) dominated the nifH sequence pool, while the remaining recovered sequences belonged to non-cyanobacterial phylotypes."*

**(7) L36 delete "where nifH sequences were recovered'**
**(8) L36-37. Rephrased as: At all the other sites nifH gene sequences were phylogenetically exclusively related to non-cyanobacterial phylotypes.**

The sentence was rephrased as follows (lines 34 and 35):

*"At all the other sites however, the recovered nifH sequences were exclusively assigned phylogenetically to non-cyanobacterial phylotypes."*

**(9) L36. Delete "We support that"**
**(10) L37. ...were likely promoted....**

We have deleted those three words, changed the sentence as suggested and it now reads as follows (lines 35 to 39):

*"The intense N₂ fixation activities recorded at the time of our study were likely promoted by the availability of phytoplankton-derived organic matter produced during the spring bloom, as evidenced by the significant surface particulate organic carbon concentrations. Also, the presence of excess phosphorus signature in surface waters seemed to contribute to sustaining N₂ fixation, particularly at the sites with extreme activities."*

**(11) L40-41. These results provide a mechanistic understanding for the unexpected high N2 fixation in productive waters of the temperate North Atlantic, and highlight the importance of N2 fixation for future assessment of global N inventory.**

We have deleted the sentence in lines 40-41 from the previous version and the above sentence was added instead.

**(12) L493. Delete able**

We have deleted that word.

**(13) L497. We speculate**

We have replaced the word "support" by "speculate" as suggested.

Regards
Zhongjun Jia

[revised manuscript text omitted]

---

## Author Response (AR3)

**Associate Editor Decision: Publish subject to technical corrections** (19 Feb 2019) by Zhongjun Jia**

Comments to the Author: Dear Fonseca-Batista

Thank you for submitting your manuscript to BG

All concerns have been addressed, and it can be published.

**Please carefully read the author guidelines of BG for format adjustment before publication**

As requested by the Editor we adapted the spacing before and after all the titles and subtitles so that our manuscript would fit exactly to lay out in the Copernicus Word Template.

Additionally, we screened the reference list, already formatted with the Copernicus Publication style on Mendeley, to make all the final adjustment, mostly by adding, when existent, the "doi" identifier when missing for some references.

Kind regards

Zhongjun

**Evidence of high N2 fixation rates in the temperate Northeast Atlantic**

**3**

4 Debany Fonseca-Batista1,2, Xuefeng Li1,3, Virginie Riou4, Valérie Michotey4, Florian Deman1,
5 François Fripiat5, Sophie Guasco4, Natacha Brion1, Nolwenn Lemaitre1,6,7, Manon Tonnard6,8,
6 Morgane Gallinari6, Hélène Planquette6, Frédéric Planchon6, Géraldine Sarthou6, Marc Elskens1,
7 Julie LaRoche2, Lei Chou3, Frank Dehairs1
8

[revised manuscript text omitted]